# Counting Distinct Elements in the Turnstile Model with Differential Privacy under Continual Observation

**Palak Jain**
Boston University
palakj@bu.edu

**Iden Kalemaj**
Boston University
ikalemaj@bu.edu

**Sofya Raskhodnikova**
Boston University
sofya@bu.edu

**Satchit Sivakumar**
Boston University
satchit@bu.edu

**Adam Smith**
Boston University
ads22@bu.edu

## Abstract

Privacy is a central challenge for systems that learn from sensitive data sets, especially when a system's outputs must be continuously updated to reflect changing data. We consider the achievable error for differentially private continual release of a basic statistic—the number of distinct items—in a stream where items may be both inserted and deleted (the *turnstile* model). With only insertions, existing algorithms have additive error just polylogarithmic in the length of the stream $T$. We uncover a much richer landscape in the turnstile model, even without considering memory restrictions. We show that every differentially private mechanism that handles insertions and deletions has *worst-case* additive error at least $T^{1/4}$ even under a relatively weak, *event-level* privacy definition. Then, we identify a parameter of the input stream, its *maximum flippancy*, that is low for natural data streams and for which we give tight parameterized error guarantees. Specifically, the maximum flippancy is the largest number of times that the contribution of a single item to the distinct elements count changes over the course of the stream. We present an *item-level* differentially private mechanism that, for all turnstile streams with maximum flippancy $w$, continually outputs the number of distinct elements with an $O(\sqrt{w} \cdot \operatorname{poly} \log T)$ additive error, without requiring prior knowledge of $w$. We prove that this is the best achievable error bound that depends only on $w$, for a large range of values of $w$. When $w$ is small, the error of our mechanism is similar to the polylogarithmic in $T$ error in the insertion-only setting, bypassing the hardness in the turnstile model.

## 1 Introduction

Machine learning algorithms are frequently run on sensitive data. In this context, a central challenge is to protect the privacy of individuals whose information is contained in the training set. Differential privacy [27] provides a rigorous framework for the design and analysis of algorithms that publish aggregate statistics, such as parameters of machine learning models, while preserving privacy. In this work, we focus on the model of differential privacy interchangeably called *continual observation* and *continual release* that was introduced by Dwork et al. [25] and Chan et al. [13] to study privacy in settings when both the data and the published statics are constantly updated. One of the most fundamental statistics about a data stream is the number of distinct elements it contains (see, e.g., the book by Leskovec et al. [47]). The problem of counting distinct elements has been widely studied, starting with the work of Flajolet and Martin [32], and has numerous applications [2, 30, 49, 42, 4], including monitoring the number of logins from distinct accounts to a streaming service, tracking

37th Conference on Neural Information Processing Systems (NeurIPS 2023).

the number of different countries represented by people in a chat room, and tracking the number of students signed up for at least one club at a university. Algorithms for this problem are also used as basic building blocks in more complicated data analyses.

We investigate the problem of privately counting the number of distinct elements under continual observation in the turnstile model, which allows both element insertions and deletions. In the continual release model, a data collector receives a sensitive dataset as a stream of inputs and produces, after receiving each input, an output that is accurate for all inputs received so far. The input stream is denoted $x$ and its length (also called the *time horizon*) is denoted $T$. The elements come from a universe $\mathcal{U}$. Each entry in the stream is an *insertion* (denoted by $+u$) or a *deletion* (denoted by $-u$) of some element $u \in \mathcal{U}$ or, alternatively, a *no-op* (denoted by $\perp$), representing that no update occurred in the current time step. More formally, for a universe $\mathcal{U}$, let $\mathcal{U}_\pm$ denote the set $\{+, -\} \times \mathcal{U} \cup \{\perp\}$ of possible stream entries. The shorthand $+u$ and $-u$ is used for the pairs $(+, u)$ and $(-, u)$. Given a vector $x$ of length $T$ and an integer $t \in [T]$, the vector $x[1:t]$ denotes the prefix of $x$ consisting of the first $t$ entries of $x$.

Next, we define the function CountDistinct in the (turnstile) continual release model.

**Definition 1.1** (Existence vector, CountDistinct)**.** *Fix a universe $\mathcal{U}$ and a time horizon $T \in \mathbb{N}$. For an element $u \in \mathcal{U}$ and a stream $x \in \mathcal{U}_\pm^T$, the* existence vector $f_u(x) \in \{0,1\}^T$ *is an indicator vector that tracks the existence of element $u$ in $x$: specifically, for each $t \in [T]$, the value $f_u(x)[t] = 1$ if and only if there are strictly more insertions than deletions of element $u$ in $x[1:t]$. The function* CountDistinct $: \mathcal{U}_\pm^T \to \mathbb{N}^T$ *returns a vector of the same length as its input, where* CountDistinct$(x)[t] = \sum_{u \in \mathcal{U}} f_u(x)[t]$ *for all $t \in [T]$.*

The focus of our investigation is the best achievable error in the continual release model for a given time horizon $T$ and privacy parameters. We study the worst-case (over all input streams and time steps $t$) additive error of privately approximating the distinct elements counts under continual release.

**Definition 1.2** (Error of an answer vector and error of a mechanism for CountDistinct)**.** *Given an answer vector $a \in \mathbb{R}^T$, the error of this vector with respect to the desired function value $f(x) \in \mathbb{R}^T$ computed on dataset $x$ is defined as $\mathsf{ERR}_f(x, a) = \|f(x) - a\|_\infty$. A mechanism for* CountDistinct *in the continual release model is $\alpha$-accurate if it outputs a vector of answers $a$ with error $\mathsf{ERR}_{\mathsf{CountDistinct}}(x, a) \leq \alpha$ with probability at least 0.99.*

Next, we discuss privacy. Originally, differential privacy [27] was defined in a setting where a data collector outputs the desired information about an entire dataset all at once. We call this the *batch model* to contrast it with continual release. In the batch model, two datasets are called *neighbors* if they differ in the data of one individual. There are two natural ways to adapt this definition to the continual release model [25, 13], depending on the desired privacy guarantees.

**Definition 1.3** (Neighboring streams)**.** *Let $x, x' \in \mathcal{U}_\pm^T$ be two streams of length $T$. Streams $x$ and $x'$ are* event-neighbors *if one can be obtained from the other by replacing a stream entry with $\perp$. Streams $x$ and $x'$ are* item-neighbors *if one can be obtained from the other by replacing a subset of stream entries pertaining to one specific element of $\mathcal{U}$ with symbol $\perp$.*

Differential privacy can be defined with respect to any notion of neighboring datasets. There are two privacy parameters: $\varepsilon > 0$ and $\delta \in [0, 1)$. An algorithm $\mathcal{A}$ is $(\varepsilon, \delta)$-*differentially private (DP)* if for all pairs of neighboring datasets $x, x'$ and all events $S$ in the output space of $\mathcal{A}$,

$$\Pr[\mathcal{A}(x) \in S] \leq e^\varepsilon \Pr[\mathcal{A}(x') \in S] + \delta.$$

The case when $\delta = 0$ is referred to as *pure* differential privacy, and the general case as *approximate* differential privacy. For event-neighboring (respectively, item-neighboring) streams $x, x' \in \mathcal{U}_\pm^T$, we say that $\mathcal{A}$ is $(\varepsilon, \delta)$-*event-level-DP* (respectively, *item-level-DP*). Item-level differential privacy imposes a more stringent requirement than event-level, since it guards against larger changes in the input stream. To contrast with the batch setting, we refer to continual release algorithms as *mechanisms*.

In the batch setting, where only CountDistinct$(x)[T]$ is released, there is an $\varepsilon$-DP algorithm for counting distinct elements with expected error $O(1/\varepsilon)$ since the function CountDistinct$(x)[T]$ has sensitivity 1—regardless of whether we consider event-level or item-level privacy. Privacy is more challenging in the continual release setting, where we aim to release a sequence of estimates, one for each time $t$, and we require that the privacy guarantee hold for the entire sequence of outputs.

| Bounds | Item-Level Privacy | Event-Level Privacy |
|---|---|---|
| Upper | $\tilde{O}\left(\min\left(\left(\sqrt{w_x}\log T + \log^3 T\right)\cdot\frac{\sqrt{\log 1/\delta}}{\varepsilon}, \frac{(T\log 1/\delta)^{1/3}}{\varepsilon^{2/3}}, T\right)\right)$ (Thm. 1.5) | |
| Lower | $\tilde{\Omega}\left(\min\left(\frac{\sqrt{w_x}}{\varepsilon}, \frac{T^{1/3}}{\varepsilon^{2/3}}, T\right)\right)$ (Thm. 1.7) | $\Omega\left(\min\left(\frac{\sqrt{w_x}}{\varepsilon}, \frac{T^{1/4}}{\varepsilon^{3/4}}, T\right)\right)$ (Thm. 1.6) |

Table 1: Summary of our results: bounds on the worst-case additive error for CountDistinct under event-level and item-level $(\varepsilon, \delta)$-differential privacy, with $\varepsilon \leq 1$ and $\delta = o(\frac{\varepsilon}{T})$. Upper bounds depend on the maximum flippancy $w_x$ of the input $x$, for every $x$. Lower bounds apply to the worst-case error of an algorithm taken over all inputs with a given maximum flippancy.

Prior work on privately estimating distinct elements in this setting considered the insertion-only model, exclusively: Bolot et al. [9] show that one can get a sequence of estimates, all of which are within additive error $poly(\log T)/\varepsilon$. Their result holds for both item-level and event-level privacy (which are essentially equivalent for counting distinct elements with only insertions). Follow-up work generalized their mechanism but, again, considered only insertions [35, 29].

We uncover a much richer landscape in the turnstile model, even without considering memory restrictions. We show that any differentially private mechanism that handles insertions and deletions has *worst-case* additive error at least $T^{1/4}$ even under *event-level* privacy, the weaker of the two privacy notions. To overcome this lower bound, we identify a property of the input stream, its *maximum flippancy*, that is low for natural data streams and for which one can give tight parameterized error guarantees. To define flippancy, recall the notion of the existence vector from Definition 1.1.

**Definition 1.4** (Flippancy). *Given a stream $x$ of length $T$ and an element $u \in \mathcal{U}$, the flippancy of $u$ in $x$, denoted by $\mathsf{flip}(u, x)$, is the number of pairs of adjacent entries in the existence vector $f_u(x)$ with different values. That is, $\mathsf{flip}(u, x) = |\{j \in [T-1] : f_u(x)[j] \neq f_u(x)[j+1]\}|$. The maximum flippancy of a stream $x$, denoted $w_x$, is $\max_{u \in \mathcal{U}} \mathsf{flip}(u, x)$.*

In other words, the maximum flippancy is the largest number of times the contribution of a single item to the distinct elements count changes over the course of the stream. We design item-level private mechanisms whose error scales with the maximum flippancy of the stream, even though the maximum flippancy is not an input to the mechanism. We show matching lower bounds for item-level privacy that hold in all parameter regimes. For a large range of the flippancy parameter, we also show a matching lower bound for event-level privacy, via a different argument. This leaves a range with an intriguing gap between item-level and event-level bounds.

## 1.1 Our results

Our results are summarized in Table 1. Our first result is a mechanism for privately approximating CountDistinct for turnstile streams. For a stream $x$ of length $T$ with maximum flippancy $w_x$, this mechanism is item-level-DP and has error $O\left(\min(\sqrt{w_x}\cdot \mathrm{polylog}\, T, T^{1/3})\right)$. Crucially, the mechanism is not given the maximum flippancy upfront.

**Theorem 1.5** (Upper bound). *For all $\varepsilon, \delta \in (0, 1]$ and sufficiently large $T \in \mathbb{N}$, there exists an $(\varepsilon, \delta)$-item-level-DP mechanism for CountDistinct that is $\alpha$-accurate for all turnstile streams $x$ of length $T$, where*

$$\alpha = \tilde{O}\left(\min\left(\left(\sqrt{w_x}\log T + \log^3 T\right)\cdot\frac{\sqrt{\log 1/\delta}}{\varepsilon}, \frac{(T\log 1/\delta)^{1/3}}{\varepsilon^{2/3}}, T\right)\right),$$

*and $w_x$ is the maximum flippancy of the stream $x$.*

Since this mechanism is item-level-DP, it is also event-level-DP with the same privacy parameters. The error it achieves is the best possible in terms of dependence only on $w_x$ for item-level privacy, and this error is nearly tight for event-level privacy. When $w_x$ is small, as is the case for many natural streams, our mechanism has error $O(\mathrm{polylog}\, T)$, similar to mechanisms for the insertion-only setting.

Theorem 1.5 can be easily extended to $\varepsilon$ bounded by any constant larger than 1. We fixed the bound to be 1 to simplify the presentation. Our mechanism has polynomial time and space complexity in the

input parameters, although it does not achieve the typically sublinear space guarantees of streaming algorithms. (See "Bounded Memory" in Section 1.4 for discussion.)

Our lower bounds on the accuracy for CountDistinct for turnstile streams are parametrized by a flippancy bound $w$, and apply for streams with maximum flippancy $w_x \leq w$. For event-level DP, our lower bound shows that for all mechanisms with error guarantees expressed solely in terms of the maximum flippancy $w_x$, time horizon $T$, and privacy parameter $\varepsilon$, our CountDistinct mechanism is asymptotically optimal for a large range of values of $w_x$, namely, for all $w_x \leq T^{1/2}$ and $w_x \geq T^{2/3}$. The best achievable error for $w_x \in (T^{1/2}, T^{2/3})$ for event-level DP remains an open problem.

**Theorem 1.6** (Event-level lower bound). *Let* $\varepsilon, \delta \in (0, 1]$, *and sufficiently large* $w, T \in \mathbb{N}$ *such that* $w \leq T$. *For all* $(\varepsilon, \delta)$*-event-level-DP mechanisms that are* $\alpha$*-accurate for* CountDistinct *on turnstile streams of length* $T$ *with maximum flippancy at most* $w$, *if* $\delta = o(\frac{\varepsilon}{T})$,

$$
\alpha = \Omega \left( \min \left( \frac{\sqrt{w}}{\varepsilon}, \frac{T^{1/4}}{\varepsilon^{3/4}}, T \right) \right).
$$

In particular, any accuracy bound for event-level algorithms depending only on $w_x$, as in Theorem 1.5, must grow as $\Omega \left( \min \left( \frac{\sqrt{w_x}}{\varepsilon}, \frac{T^{1/4}}{\varepsilon^{3/4}}, T \right) \right)$, in all ranges of $w_x$. This is reflected in Table 1.

For item-level DP, our lower bound on the error matches our upper bound for all regimes of $w_x$ up to polylogarithmic factors.

**Theorem 1.7** (Item-level lower bound). *Let* $\varepsilon \in (0, 1]$, $\delta \in (0, 1]$, *and sufficiently large* $w, T \in \mathbb{N}$ *such that* $w \leq T$. *For all* $(\varepsilon, \delta)$*-item-level-DP mechanisms that are* $\alpha$*-accurate for* CountDistinct *on turnstile streams of length* $T$ *with maximum flippancy at most* $w$,

   **1** *If* $\delta = o(\varepsilon/T)$, *then* $\alpha = \tilde{\Omega} \left( \min \left( \frac{\sqrt{w}}{\varepsilon}, \frac{T^{1/3}}{\varepsilon^{2/3}}, T \right) \right)$.

   **2** *If* $\delta = 0$, *then* $\alpha = \Omega \left( \min \left( \frac{w}{\varepsilon}, \sqrt{\frac{T}{\varepsilon}}, T \right) \right)$.

In particular, any accuracy bounds depending only on $w_x$ must grow at least as quickly as the expressions in Table 1.

**Variants of the model.** All our lower bounds also hold in the *strict turnstile model*, where element counts never go below 0. They also apply to *offline* mechanisms that receive the entire input stream before producing output; they do not rely on the mechanism's uncertainty about what comes later in the stream. Furthermore, our item-level lower bounds hold even in the model where each element can be inserted only when it is absent and deleted only when it is present (as is the case, for example, with the "like" counts on social media websites).

## 1.2 Our techniques

**Upper bound techniques: tracking the maximum flippancy.** Before describing our algorithmic ideas, we explain the main obstacle to using the techniques previously developed for insertion-only streams [9, 29] in the turnstile setting. Bolot et al. [9] and Epasto et al. [29] used a reduction from CountDistinct to the summation problem. A mechanism for the summation problem outputs, at every time step $t \in [T]$, the sum of the first $t$ elements of the stream. Dwork et al. [27] and Chan et al. [13] designed the binary-tree mechanism to obtain a $O(\text{polylog } T)$-accurate mechanism for summation. Given an input stream $x$ of length $T$ (to the CountDistinct problem), define a corresponding summation stream $s_x \in \{-1, 0, 1\}^T$. At time step $t \in [T]$, the entry $s_x[t]$ equals the difference in the count of distinct elements between time steps $t - 1$ and $t$, i.e., $s_x[t] = \text{CountDistinct}(x)[t] - \text{CountDistinct}(x)[t-1]$. Then CountDistinct$(x)[t]$ is precisely the sum of the first $t$ elements of $s_x$. In the insertion-only model, changing one entry of $x$ changes at most 2 entries of $s_x$, and thus, by group privacy, the binary-tree mechanism has $O(\text{polylog } T)$ additive error for CountDistinct. For turnstile streams, even under the weaker notion of event-level privacy, a change in the stream $x$ can cause $\Omega(T)$ changes to $s_x$. To see this, consider the stream consisting of alternating insertions $(+u)$ and deletions $(-u)$ of a single element $u \in \mathcal{U}$, and its event-neighboring stream where the first occurrence of $+u$ is replaced with $\perp$. This example illustrates that one of the difficulties of the CountDistinct problem for turnstile streams lies with items that switch from being present to absent multiple times over the course of the stream, that is, have high flippancy. We present

a private mechanism that outputs estimates of the count of distinct elements in a turnstile stream with optimal accuracy in terms of maximum flippancy.

Our first key idea allows us to obtain a mechanism, Algorithm 1, that is given as input a flippancy upper bound $w$. For a stream $x$ whose maximum flippancy is bounded by $w$, changing to an item-neighbor of $x$ causes at most $2w$ changes to the corresponding summation stream $s_x$. This observation, combined with a group privacy argument, gives a mechanism with error $O(w \cdot \text{polylog } T)$ directly from the accuracy guarantee of the binary-tree mechanism for summation. Previous works in the insertion-only model [9, 29] used precisely this approach for the special case $w = 1$. To obtain the better $\sqrt{w}$ dependence on $w$ in our upper bound, we "open up" the analysis of the binary-tree mechanism. By examining the information stored in each node of the binary tree for the summation stream, we show that changing the occurrences of one item in a stream $x$ with maximum flippancy at most $w$ can change the values of at most $w$ nodes in each *level* of the binary tree. The $\sqrt{w}$ dependence in the error then follows from the privacy guarantees of the Gaussian mechanism (used inside the binary-tree mechanism) for approximate differential privacy. This type of noise reduction makes crucial use of the binary tree approach: there are optimized noise addition schemes for prefix sums that improve quantitatively over the binary-tree mechanism (see, e.g., [18, 41]), but it is unclear if they allow the same noise reduction. While our mechanism is only accurate for streams with maximum flippancy at most $w$, it is private even for streams that violate this condition. To achieve this, our mechanism ignores stream elements after their flippancy exceeds $w$.

The second key idea allows our algorithms to adapt automatically to the maximum flippancy $w_x$ of the input, without the need for an a-priori bound $w$. We design a private mechanism, Algorithm 3, that approximately keeps track of the maximum flippancy of the prefix of the stream seen so far and invokes our first mechanism (Algorithm 1) with the current estimated maximum flippancy $\hat{w}$ as an input. Our main innovation lies in the careful application of the sparse vector algorithm [24] to track the maximum flippancy of the stream. We cannot do this directly, since the sparse vector algorithm achieves good utility only for queries of low sensitivity, and maximum flippancy has global sensitivity $\Omega(T)$ under item-level changes.

Instead, we track a low sensitivity proxy that indirectly monitors the maximum flippancy $w_x$: given the current estimate $\hat{w}$ of the flippancy, we use the sparse vector algorithm to continuously query *the number of items in the stream with flippancy above $\hat{w}$*. This query has sensitivity one for item-level neighbors, as desired, but it is not a priori clear how to use it to upper bound the maximum flippancy of the stream. This is remedied by observing that Algorithm 1, invoked with a flippancy bound $\hat{w}$, has the same error (and privacy) guarantee even if at most $\sqrt{\hat{w}}$ items in the stream have flippancy higher than $\hat{w}$. That is, an exact upper bound on the maximum flippancy is not needed to design an accurate mechanism. Items that violate the flippancy bound are safely ignored by Algorithm 1 and do not contribute to the distinct elements count.

When the number of high-flippancy items gets large, we adjust the estimate $\hat{w}$ and invoke a new instantiation of Algorithm 1. By doubling $\hat{w}$ each time this happens, we ensure that it remains at most twice the actual maximum flippancy $w_x$, and that we need only invoke $\log T$ different copies of Algorithm 1 and the sparse vector algorithm[1]. With these ideas, we obtain an item-level private mechanism that, for all streams $x$, has error that scales with $\sqrt{w_x}$.

**Lower bound techniques.** Our lower bounds use the embedding technique introduced by Jain et al. [44] to obtain strong separations between the batch and continual release models of differential privacy. The approach of Jain et al. embeds multiple separate instances of an appropriately chosen base problem *on the same sensitive dataset* in the batch model into a single instance of a continual release problem. Then, the continual release mechanism can be used to solve multiple instances of the base problem in the batch model. The hardness results in the continual release model follow from lower bounds for the batch model.

A key idea in our event-level lower bound is a connection between the inner product of two binary vectors and the count of distinct elements in the union of those indices where the vector bits equal 1. Estimates of distinct elements counts can thus be used to estimate inner products on a sensitive dataset of binary bits. Lower bounds on the accuracy of private algorithms for estimating inner product queries have been previously established in the batch model through the reconstruction attack of Dinur and Nissim [21]. This connection was used by Mir et al. [50] to provide lower bounds for

---

[1]All log expressions in this article are base 2.

pan-private algorithms for counting distinct elements. However, continual release and pan-privacy are orthogonal notions, and their results don't imply any lower bounds in our setting. We crucially use deletions to embed multiple instances of inner product queries into a stream: once a query is embedded and the desired estimate is received, the elements inserted to answer that query can be entirely deleted from the stream to obtain a "clean slate" for the next query. We obtain a lower bound of $T^{1/4}$ on the error of event-level private mechanisms for CountDistinct in turnstile streams.

We obtain our stronger item-level lower bounds (for pure and approximate differential privacy) by embedding multiple instances of a 1-way marginal query. We then apply lower bounds of Hardt and Talwar [38] and Bun et al. [12] for releasing all 1-way marginals in the batch model in conjunction with our reduction. The 1-way marginals of a dataset $y \in \{0,1\}^{n \times d}$, consisting of $n$ records and $d$ attributes, are the averages of all $d$ attributes of $y$. Deletions in the stream are once again crucially used to embed a marginal query for one attribute and then clean the slate for the next attribute. Changing one record/row in the dataset $y$ translates to $d$ changes of an item in the constructed stream, and thus this reduction is particularly tailored to item-level lower bounds.

## 1.3 Related work

The study of differential privacy under continual release was initiated by two concurrent works [25, 13]. They proposed the binary-tree mechanism for computing sums of binary bits. The binary-tree mechanism has found numerous applications in the continual release setting and elsewhere, demonstrating the versatility of this mechanism. Under continual release, it has been extended to work for sums of real values [52], weighted sums [9], graph statistics [55, 31], and most relevantly, counting distinct elements [9, 29, 35]. It has also been employed for private online learning [45, 57, 1] and for answering range queries [25, 26, 28].

Prior to our work, the CountDistinct problem with continual release was studied exclusively in the insertions-only model. Bolot et al. [9] were the first to study this problem and showed a $O(\log^{1.5} T)$-accurate item-level-DP mechanism. Ghazi et al. [35] considered the more challenging sliding-window model and showed nearly-matching upper and lower bounds for this setting, parameterized by the window size, for item-level and event-level differential privacy. Epasto et al. [29] studied the more general $\ell_p$-frequency estimation problem with a focus on space efficiency. For distinct elements, i.e., $p = 0$, their mechanism provides an estimate with $1 + \eta$ multiplicative error and $O(\log^2 T)$ additive error, using space $\text{poly}(\log T/\eta)$. They also extended their results to the sliding-window model. Two of the works [9, 29] reduced the CountDistinct problem to the bit summation primitive, which allowed them to use the binary-tree mechanism. Since the streams are restricted to be insertion-only, the bit summation primitives they considered have low constant sensitivity. The same primitives have sensitivity $\Omega(T)$ for turnstile streams, and thus this approach cannot be directly extended to our setting. Ghazi et al. [35] observed that for fixed and sliding windows, the distinct elements problem can be reduced to range queries. For the special case when the window is the entire stream, their reduction is to the summation problem.

In concurrent work, Henzinger et al. [40] studied CountDistinct with insertions and deletions in a different version of the continual release model (which we call the 'likes' model), where an item can be deleted at a time step only if it is already present in the stream at that time step, and inserted only if it is absent from the stream at that time step. Our model is more general, since multiple insertions and deletions of the same item can happen consecutively. Our upper bound and our item-level privacy lower bound can be extended to the 'likes' model. On the other hand, our event-level private lower bound provably does not apply to that model: in the 'likes' model, for event-level privacy, there is a simple reduction to the bit summation problem in the continual release model such that the resulting algorithm incurs only a polylogarithmic in $T$ error, whereas we show that in our model, any event-level private algorithm incurs a polynomial in $T$ error.

Henzinger et al. [40] showed error bounds for item-level privacy in the likes model that are parameterized by the total number of updates $K$ in the stream. The parameter $K$ is related to our concept of flippancy: in the likes model, $K$ equals the sum of all items' flippancies and, in general, is at least that sum. Henzinger et al. [40] give an $(\varepsilon, 0)$-DP algorithm with error $\tilde{O}(\sqrt{K} \log T)$ and show a nearly matching lower bound on the error for $(\varepsilon, 0)$-DP algorithms using a packing argument. This lower bound applies to our model as well. It is incomparable to our lower bounds, since it scales differently and depends on a different parameter. In our model, their algorithm can be analyzed to give

error bounds in terms of the sum $K'$ of the flippancies of the items and incurs error $\tilde{O}(\sqrt{K'}\log T)$; however, it is unclear if their algorithm can be analyzed in our model to give bounds in terms of the (potentially smaller) maximum flippancy.

Another line of work investigated private sketches for distinct elements, motivated by the popularity of sketching algorithms for the streaming setting. Mergeable sketches for counting distinct elements have received particular attention [56, 15, 51, 39], since they allow multiple parties to estimate the joint count of distinct elements by merging their private sketches. While these sketches can be combined with the binary-tree mechanism to obtain private mechanisms for CountDistinct, the utility deteriorates when many $(\log T)$ sketches are merged. In fact, Desfontaines et al. [19] showed that achieving both privacy and high accuracy is impossible when many sketches for counting distinct elements are merged. Other private sketches have been studied [54, 20, 58] for the streaming batch setting (without continual release). The distinct elements problem has also been studied in a distributed setting [14, 34] and under pan-privacy [50]. In particular, our lower bound for event-level privacy uses ideas from the lower bound of Mir et al. [50], as described in Section 1.2

The CountDistinct problem has been extensively studied in the non-private streaming setting, where the goal is to achieve low space complexity [32, 3, 16, 36, 37, 6, 5, 22, 43, 59, 30, 7, 33, 10, 46]. Blocki et al. [8] showed a black-box transformation for every streaming algorithm with tunable accuracy guarantees into a DP algorithm with similar accuracy, for low sensitivity functions. Their transformation does not obviously extend to the continual release setting. Moreover CountDistinct has high sensitivity for turnstile streams.

The first lower bound in the continual release model of differential privacy was an $\Omega(\log T)$ bound on the accuracy of mechanisms for bit summation, shown by Dwork et al. [25]. Jain et al. [44] gave the first polynomial separation in terms of error between the continual release model and the batch model under differential privacy. Our lower bounds also show such a separation. The lower bounds of Jain et al. [44] were for the problems of outputting the value and index of the attribute with the highest sum, amongst $d$ attributes of a dataset. Our lower bounds are inspired by their sequential embedding technique to reduce multiple instances of a batch problem to a problem in the continual release model. Similar to them, we also reduce from the 1-way marginals problem to obtain our item-level lower bound. However, our event-level lower bound involves reducing from a different problem, and our reductions use the specific structure of CountDistinct for turnstile streams.

## 1.4 Broader impact, limitations, and open questions

We study the achievable error of DP mechanisms for CountDistinct under continual observation in streams with insertions and deletions. We show that it is characterized by the *maximum flippancy* of the stream. Our work is motivated by societal concerns, but focused on fundamental theoretical limits. It contributes to the broader agenda of obtaining privacy-preserving algorithms for data analysis. We discuss natural directions for future research and some limitations of our work.

**Tight bounds:** We found the best achievable error in some settings, but our upper and lower bounds do not match in some parameter regimes. What is the right error bound for event-level privacy for streams $x$ with maximum flippancy $w_x$ between $\sqrt{T}$ and $T^{2/3}$? Our results yield a lower bound of $T^{1/4}$ and an upper bound of roughly $\sqrt{w_x}$.

**Bounded memory:** We did not consider any memory restrictions. Prior to our work, no other work addressed CountDistinct with deletions under continual release—with or without space constraints. We consider only the privacy constraint since it is more fundamental—it cannot be avoided by buying more memory—and the best algorithms with unbounded memory provide a benchmark by which to evaluate space-constrained approaches.

Space complexity is certainly a natural topic for future work. While it is not clear how to apply the sketching techniques of Epasto et al. [29] to the turnstile setting, it would be interesting to come up with accurate, private, and low-memory mechanisms for counting distinct elements in turnstile streams. Such algorithms would necessarily mix multiplicative and additive error guarantees (due to space and privacy constraints, respectively).

## 1.5 Organization

In Section 2, we present a mechanism for privately approximating CountDistinct on turnstile streams. Preliminaries on differential privacy, proofs omitted from Section 2 as well as proofs of Theorems 1.6 and 1.7 all appear in the supplementary material.

## 2 Item-level private mechanisms for CountDistinct

In this section, we present our item-level-DP mechanism for CountDistinct for turnstile streams. Its guarantees are stated in Theorem 2.1 with zero Concentrated Differential Privacy (zCDP). Using this notion of privacy, one can show tight bounds for the Gaussian mechanism and cleaner and tighter bounds for composition. Theorem 2.1 is the key ingredient in our proof of Theorem 1.5. This section is dedicated to our CountDistinct mechanism and the main ideas behind it. All proofs are deferred to the supplementary material.

**Theorem 2.1** (Upper bound). *For all $\rho \in (0, 1]$ and sufficiently large $T \in \mathbb{N}$, there exists a $\rho$-item-level-zCDP mechanism for* CountDistinct *that is $\alpha$-accurate for all turnstile streams $x$ of length $T$, where*

$$\alpha = O\Big(\frac{\sqrt{w_x}\log T + \log^3 T}{\sqrt{\rho}}\Big),$$

*and $w_x$ is the maximum flippancy of the stream $x$.*

In Section 2.1, we describe a modification to the binary-tree mechanism which, when analyzed carefully, provides the desired error guarantees—but only if the maximum flippancy of the stream is known upfront. In Section 2.2, we use this mechanism, in conjunction with a method for adaptively estimating the flippancy bound, to obtain our item-level-DP mechanism for CountDistinct.

## 2.1 Enforcing a given flippancy bound $w$

When a flippancy upper bound $w$ is given upfront, we leverage the structure of the binary-tree mechanism to privately output the number of distinct elements at each time $t \in [T]$, where $T$ is the stream length. The mechanism and its error guarantees are presented in Algorithm 1 and Theorem 2.2, respectively. To provide intuition, we first describe the mechanism when it is run on streams with maximum flippancy at most $w$. We then discuss a modification that ensures privacy of the mechanism for all streams regardless of maximum flippancy.

Algorithm 1 stores vectors $\tilde{f}_u \in \{0, 1\}^T$ for all elements $u \in \mathcal{U}$ that appear in the stream. For streams with maximum flippancy at most $w$, the vector $\tilde{f}_u$ is equal to the existence vector $f_u$. In this case, by Definition 1.1, the number of distinct elements at timestep $t \in [T]$ equals $\sum_{u \in \mathcal{U}} \tilde{f}_u[t]$. The mechanism outputs values $\sum_{u \in \mathcal{U}} \tilde{f}_u[t]$ with Gaussian noise added according to the binary-tree mechanism, with privacy parameter $\approx \rho/w$ (see Definition 2.4)—that is, with noise scaled up by a factor of $\approx \sqrt{w/\rho}$. The accuracy of this mechanism follows from that of the binary-tree mechanism.

However, if the mechanism computed $f_u$ instead of $\tilde{f}_u$, it would not be private for streams with maximum flippancy greater than $w$, since it adds noise that scales according to $w$. That is because for every stream $x \in \mathcal{U}_{\pm}^T$ with maximum flippancy $w_x > w$ there exists a neighboring stream $x'$ such that the vectors CountDistinct$(x)$ and CountDistinct$(x')$ differ in as many as $\Theta(w_x)$ indices. To provide privacy for such streams, the mechanism simply "truncates" the vector $f_u \in \{0, 1\}^T$ to obtain $\tilde{f}_u[t] = 0$ for all $t \geq t^*$ if the flippancy of $u$ in $x[1 : t^*]$ exceeds $w$. This corresponds to running the naive version of the mechanism (that uses $f_u$ instead of $\tilde{f}_u$) on the "truncated" version of the stream $x$, where elements in $x$ are ignored after their flippancy exceeds $w$. (Note that the computation of $\tilde{f}_u$ can be done online since $\tilde{f}_u[t]$ depends only on $x[1 : t]$.) With a careful analysis of the value stored in each node of the binary tree, we are able to show that this mechanism is $\rho$-item-level-zCDP for all streams, however, it loses accuracy for streams with many high flippancy elements. In Section 2.2, we leverage this mechanism to provide estimates of CountDistinct that are both accurate and private for *all* streams.

**Theorem 2.2** (Mechanism for a given flippancy bound $w$). *Fix $\rho \in (0, 1]$, sufficiently large $T \in \mathbb{N}$, and $w \leq T$. Algorithm 1 is a mechanism for* CountDistinct *for turnstile streams that is $\rho$-item-*

*level-zCDP for all input streams of length $T$, and $\alpha$-accurate for streams of length $T$ with maximum flippancy at most $w$, where $\alpha = O\left(\frac{\sqrt{w}\log T + \log^3 T}{\sqrt{\rho}}\right)$.*

---

**Algorithm 1** Mechanism $\mathcal{M}$ for CountDistinct with given flippancy bound

---

    **Input:** Time horizon $T \in \mathbb{N}$, privacy parameter $\rho > 0$, flippancy bound $w > 0$, stream $x \in \mathcal{U}_{\pm}^T$
    **Output:** Vector $s \in \mathbb{R}^T$ of distinct count estimates
1: Sample a binary-tree random variable $Z \in \mathbb{R}^T$ with parameter $\rho' = \frac{\rho}{4w(\log T + 1)}$   ▷ Definition 2.4
2: Initialize $\mathcal{U}_x = \emptyset$
3: **for all** $t \in [T]$ **do**
4:     Obtain entry $x[t]$ and skip to Step 10 if $x[t] = \perp$
5:     Suppose $x[t]$ is an insertion or deletion of a universe element $u$
6:     **if** $u \notin \mathcal{U}_x$ **then** insert $u$ into $\mathcal{U}_x$; initialize $\text{count}_u = 0$ and $\tilde{f}_u = 0^T$    ▷ vector with $T$ zeros
7:     **if** $x[t] = +u$ **then** $\text{count}_u += 1$ **else** $\text{count}_u -= 1$
8:     **for all** $v \in \mathcal{U}_x$ **do**
9:         **if** $\text{flip}(v, x[1:t]) \leq w$ and $\text{count}_v > 0$ **then** set $\tilde{f}_v[t] = 1$
10:     Return $s[t] = (\sum_{u \in \mathcal{U}_x} \tilde{f}_u[t]) + Z[t]$

---

**Definition 2.3** (Dyadic decomposition). *For $t \in \mathbb{N}$, the dyadic decomposition of the interval $(0, t]$ is a set of at most $\log t + 1$ disjoint intervals whose union is $(0, t]$, obtained as follows. Consider the binary representation of $t$ (which has at most $\log t + 1$ bits), and express $t$ as a sum of distinct powers of $2$. Then, the first interval is $(0, r]$, where $r$ is the largest power of $2$ in the sum. The second interval starts at $r + 1$ and its size is the second largest power of $2$ in the sum. The remaining intervals are defined similarly for all remaining summands. For example, for $t = 11 = 8 + 2 + 1$, the dyadic decomposition of $(0, 11]$ is the intervals $(0, 8]$, $(8, 10]$ and $(10, 11]$.*

**Definition 2.4** (Binary tree and binary-tree random variable). *Let $\rho > 0$ be a privacy parameter and $T \in \mathbb{N}$ be a power of $2$. Consider a complete binary tree with $T$ leaves whose nodes are labeled as follows. The $T$ leaves are labeled by the intervals $(t - 1, t]$ for all $t \in [T]$ and the internal nodes are labeled by intervals obtained from the union of their children's intervals. Specifically, the binary tree consists of $\log T + 1$ levels. A level $\ell \in [0, \log T]$ partitions the interval $(0, T]$ into a set of $\frac{T}{2^\ell}$ disjoint intervals, each of length $2^\ell$, of the form $((i - 1) \cdot 2^\ell, i \cdot 2^\ell]$. The nodes in level $\ell$ are labelled by the intervals in this partition.*

*The binary-tree random variable $Z \in \mathbb{R}^T$ with parameter $\rho$ is defined as follows. For each node $(t_1, t_2]$ in the binary tree with $T$ leaves, let $Z_{(t_1, t_2]} \sim \mathcal{N}(0, 1/\rho)$. For each $t \in [T]$, consider the dyadic decomposition of the interval $(0, t]$ (Definition 2.3) and let $Z[t]$ be the sum of the random variables corresponding to the intervals in this dyadic decomposition.*

The proof of Theorem 2.2 can be found in Section B.

## 2.2 Adaptively estimating a good flippancy bound $w$

In this section, we leverage the privacy and accuracy guarantees of Algorithm 1 to construct a new mechanism (Algorithm 3) for estimating CountDistinct. It achieves the privacy and accuracy guarantees of Theorem 2.1, when the maximum flippancy is not known upfront. Algorithm 3 instantiates $\log T + 1$ different copies $\mathcal{B}_0, \ldots \mathcal{B}_{\log T}$ of Algorithm 1 with flippancy bounds $2^0, \ldots, 2^{\log T}$, respectively (the maximum flippancy of a stream is at most $T$). To obtain an accurate estimate of the distinct elements count, at each time $t \in [T]$, we privately select $i \in [0, \log T]$ such that the output of $\mathcal{B}_i$ satisfies the desired accuracy guarantee for the stream entries $x[1:t]$ received so far. Selecting such $i$ amounts to selecting a good bound on the maximum flippancy of the stream $x[1:t]$. Next, we describe how to obtain this bound using the sparse vector technique (Algorithm 2).

The maximum flippancy has high sensitivity; changing one stream entry can change the maximum flippancy drastically. However, the number of items with flippancy greater than any particular threshold is a function of sensitivity one. Furthermore, since Algorithm 1 when run with flippancy bound $w$ already has error about $\sqrt{w/\rho}$, its accuracy guarantee remains asymptotically the same even if it simply ignores that many elements with flippancy greater than $w$. Thus, Algorithm 3 uses the sparse vector technique to maintain an upper bound on the flippancy of $x[1:t]$ such that not too

many elements in $x[1:t]$ violate that bound. This bound, in combination with the error guarantee of Algorithm 1, suffices to provide the desired low error guarantee. Since the sparse vector algorithm remains differentially private even when its queries are chosen adaptively, the privacy guarantees of Algorithm 3 follow from the privacy of Algorithms 1 and 2.

---

**Algorithm 2** SVT: Answering Threshold Queries with Sparse Vector Technique

---

    **Input:** Stream $x$, queries $q_1, q_2, \ldots$ of sensitivity 1, cutoff $c > 0$, privacy parameter $\rho$
    **Output:** Stream of Above or Below answers
1: Let $\varepsilon = \sqrt{2\rho}$ and set count $= 0$
2: Let $Z \sim \text{Lap}(2/\varepsilon)$
3: **for** each query $q_t$ **do**
4:     Let $Z_t \sim \text{Lap}(4c/\varepsilon)$
5:     **if** $q_t(x) + Z_t \geq Z$ and count $< c$ **then**
6:         Return Above
7:         count = count $+ 1$
8:     **else**
9:         Return Below

---

---

**Algorithm 3** Mechanism $\mathcal{M}$ for CountDistinct

---

    **Input:** Time horizon $T \in \mathbb{N}$, privacy parameter $\rho > 0$, stream $x \in \mathcal{U}_\pm^T$
    **Output:** Vector $s$ of distinct count estimates
1: Initialize vector $w_{\max} = 1 \circ 0^{T-1}$
2: **for all** $i \in [0, \log T]$ **do**
3:     Initialize $\mathcal{B}_i$ as Algorithm 1 with horizon $T$, privacy parameter $\frac{\rho}{2(\log T + 1)}$, flippancy $2^i$
4: Initialize SVT with privacy parameter $\rho/2$ and cutoff $\log T$           $\triangleright$ See Algorithm 2
5: **for all** $t \in [T]$ **do**
6:     Obtain entry $x[t]$
7:     If $t \geq 2$, set $w_{\max}[t] = w_{\max}[t-1]$
8:     **for all** $i \in [\log T]$ **do**
9:         Send $x[t]$ to mechanism $\mathcal{B}_i$ and get output $s_{i,t}$
10:     **while** True **do**
11:         Consider query $q_t = |\{u \in \mathcal{U} : \text{flip}(u, x[1:t]) \geq w_{\max}[t]\}| - \sqrt{\frac{w_{\max}[t]}{\rho}}$
12:         Send query $q_t$ to SVT and if the output is "Below", **break**
13:         Update $w_{\max}[t] = 2 \cdot w_{\max}[t]$
14:     Return $s_{j,t}$ for $j = \log(w_{\max}[t])$           $\triangleright$ Note that $j \in [0, \log T]$

---

The proof of Theorem 2.1 can be found in Section C.

## Acknowledgments and Disclosure of Funding

We thank Teresa Steiner and an anonymous reviewer for useful suggestions on the initial version of this paper. S.S. was supported by NSF award CNS-2046425 and Cooperative Agreement CB20ADR0160001 with the Census Bureau. A.S. and P.J. were supported in part by NSF awards CCF-1763786 and CNS-2120667 as well as Faculty Awards from Google and Apple.

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
