# Supplementary Material

## A  Additional background on differential privacy

In this section, we describe basic results on differential privacy used to obtain our theorems.

We denote by $\mathcal{N}(\mu, \sigma^2)$ the Gaussian distribution with mean $\mu$ and standard deviation $\sigma$. The Laplace distribution with mean 0 and standard deviation $\sqrt{2}b$ is denoted by $\mathrm{Lap}(b)$.

**Definition A.1** (($\varepsilon, \delta$)-indistinguishability). *Two random-variables $R_1, R_2$ over the same outcome space $\mathcal{Y}$ (and $\sigma$-algebra $\Sigma_{\mathcal{Y}}$) are ($\varepsilon, \delta$)-indistinguishable, denoted $R_1 \approx_{(\varepsilon, \delta)} R_2$, if for all events $S \in \Sigma_{\mathcal{Y}}$, the following hold:*

$$\Pr[R_1 \in S] \le e^{\varepsilon} \Pr[R_2 \in S] + \delta;$$
$$\Pr[R_2 \in S] \le e^{\varepsilon} \Pr[R_1 \in S] + \delta.$$

**Definition A.2** ($k$-Neighboring streams). *Let $x, x' \in \mathcal{U}_{\pm}^{T}$ be two streams of length $T$. For any natural number $k$, streams $x$ and $x'$ are $k$-event-neighbors if one can be obtained from the other in $k$ operations, where each operation involves replacing a stream entry with $\perp$ or vice-versa. For any natural number $k$, streams $x$ and $x'$ are $k$-item-neighbors if one can be obtained from the other in $k$ operations, where each operation involves replacing a subset of stream entries pertaining to one specific element of $\mathcal{U}$ with symbol $\perp$, or vice versa. 1-item-neighbors and 1-event-neighbors are called item-neighbors and event-neighbors, respectively.*

**Lemma A.3** (Group privacy [27]). *Every ($\varepsilon, \delta$)-event-level (or item-level) DP mechanism $\mathcal{M}$ is ($\ell\varepsilon, \delta'$)-event-level (or item-level) DP for groups of size $\ell$, where $\delta' = \delta \frac{e^{\ell\varepsilon} - 1}{e^{\varepsilon} - 1}$; that is, for all $\ell$-event (or $\ell$-item) neighboring data streams $x, x'$, it holds that $\mathcal{M}(x) \approx_{\ell\varepsilon, \delta'} \mathcal{M}(x')$.*

## A.1 Preliminaries on zero-concentrated differential privacy (zCDP)

This section describes *zero-concentrated differential privacy (zCDP)*, a variant of differential privacy that is less stringent than pure differential privacy, but more stringent than approximate differential privacy. Using this notion of privacy, one can show tight bounds for the Gaussian mechanism and cleaner and tighter bounds for composition. In contrast to $(\varepsilon, \delta)$-differential privacy, zCDP requires output distributions on all pairs of neighboring datasets to be $\rho$-close (Definition A.5) instead of $(\varepsilon, \delta)$-indistinguishable.

**Definition A.4** (Rényi divergence [53]). *Let $Q$ and $Q'$ be distributions on $\mathcal{Y}$. For $\xi \in (1, \infty)$, the Rényi divergence of order $\xi$ between $Q$ and $Q'$ (also called the $\xi$-Rényi Divergence) is defined as*

$$D_\xi(Q \| Q') = \frac{1}{\xi - 1} \log \left( \mathbb{E}_{r \sim Q'} \left[ \left( \frac{Q(r)}{Q'(r)} \right)^{\xi - 1} \right] \right). \tag{1}$$

*Here $Q(\cdot)$ and $Q'(\cdot)$ denote either probability masses (in the discrete case) or probability densities (when they exist). More generally, one can replace $\frac{Q(.)}{Q'(.)}$ with the the Radon-Nikodym derivative of $Q$ with respect to $Q'$.*

**Definition A.5** ($\rho$-Closeness). *Random variables $R_1$ and $R_2$ over the same outcome space $\mathcal{Y}$ are $\rho$-close (denoted $R_1 \simeq_\rho R_2$) if for all $\xi \in (1, \infty)$,*

$$D_\xi(R_1 \| R_2) \leq \xi \rho \text{ and } D_\xi(R_2 \| R_1) \leq \xi \rho,$$

*where $D_\xi(R_1 \| R_2)$ is the $\xi$-Rényi divergence between the distributions of $R_1$ and $R_2$.*

**Definition A.6** (zCDP in batch model [11]). *A randomized batch algorithm $\mathcal{A} : \mathcal{X}^n \to \mathcal{Y}$ is $\rho$-zero-concentrated differentially private ($\rho$-zCDP), if, for all neighboring datasets $y, y' \in \mathcal{X}^n$,*

$$\mathcal{A}(y) \simeq_\rho \mathcal{A}(y').$$

One major benefit of using zCDP is that this definition of privacy admits a clean composition result. We use it when analysing the privacy of the algorithms in Section 2.

**Lemma A.7** (Composition [11]). *Let $\mathcal{A} : \mathcal{X}^n \to \mathcal{Y}$ and $\mathcal{A}' : \mathcal{X}^n \times \mathcal{Y} \to \mathcal{Z}$ be batch algorithms. Suppose $\mathcal{A}$ is $\rho$-zCDP and $\mathcal{A}'$ is $\rho'$-zCDP. Define batch algorithm $\mathcal{A}'' : \mathcal{X}^n \to \mathcal{Y} \times \mathcal{Z}$ by $\mathcal{A}''(y) = \mathcal{A}'(y, \mathcal{A}(y))$. Then $\mathcal{A}''$ is $(\rho + \rho')$-zCDP.*

**Lemma A.8** (Post-processing [27, 11]). *If $\mathcal{A} : \mathcal{Y} \to \mathbb{R}^k$ is $(\varepsilon, \delta)$-DP and $\mathcal{B} : \mathbb{R}^k \to \mathcal{Z}$ is any randomized function, then the algorithm $\mathcal{B} \circ \mathcal{A}$ is $(\varepsilon, \delta)$-DP. Similarly, if $\mathcal{A}$ is $\rho$-zCDP then the algorithm $\mathcal{B} \circ \mathcal{A}$ is $\rho$-zCDP.*

The *Gaussian mechanism*, defined next, is used in Section 2. It privately estimates a real-valued function on a database by adding Gaussian noise to the value of the function.

**Definition A.9** (Sensitivity). *Let $f : \mathcal{Y} \to \mathbb{R}^k$ be a function. Its $\ell_2$-sensitivity is defined as*

$$\max_{neighbors \ y, y' \in \mathcal{Y}} \| f(y) - f(y') \|_2.$$

*To define $\ell_1$-sensitivity, we replace the $\ell_2$ norm with the $\ell_1$ norm.*

**Lemma A.10** (Gaussian mechanism [11]). *Let $f : \mathcal{X}^n \to \mathbb{R}$ be a function with $\ell_2$-sensitivity at most $\Delta_2$. Let $\mathcal{A}$ be the batch algorithm that, on input $y$, releases a sample from $\mathcal{N}(f(y), \sigma^2)$. Then $\mathcal{A}$ is $(\Delta_2^2/(2\sigma^2))$-zCDP.*

The final lemma in this section relates zero-concentrated differential privacy to $(\varepsilon, \delta)$-differential privacy.

**Lemma A.11** (Conversion from zCDP to DP [11]). *For all $\rho, \delta > 0$, if batch algorithm $\mathcal{A}$ is $\rho$-zCDP, then $\mathcal{A}$ is $(\rho + 2\sqrt{\rho \log(1/\delta)}, \delta)$-DP. Conversely, if $\mathcal{A}$ is $\varepsilon$-DP, then $\mathcal{A}$ is $(\frac{1}{2}\varepsilon^2)$-zCDP.*

## B  Proofs omitted from Section 2.1

In this section, we prove Theorem 2.2 by formalizing the ideas described at the start of Section 2.1.

*Proof of Theorem 2.2.* We start by reasoning about the privacy of Algorithm 1. It is helpful to think about Algorithm 1 more explicitly in terms of the binary tree mechanism. We define a mechanism $\mathcal{M}'$ that returns noisy values for all nodes of the binary tree from Definition 2.4 and show that the output of Algorithm 1 can be obtained by post-processing the output of $\mathcal{M}'$.

Assume w.l.o.g. that $T$ is a power of 2; otherwise, consider the value $2^{\lceil \log_2 T \rceil}$ instead. Fix a stream $x$ as the input to Algorithm 1. For all $t \in [T]$, let $F[t] = \sum_{u \in \mathcal{U}} \tilde{f}_u[t]$, where the vector $\tilde{f}_u$ is obtained by the end of running Algorithm 1 with input $x$. (If $u \notin \mathcal{U}_x$, set $\tilde{f}_u = 0^T$. Set $F[0] = 0$). Define $\mathcal{M}'$ so that on input $x$, for each node $(t_1, t_2]$ of the binary tree with $T$ leaves, it outputs $F[t_2] - F[t_1] + Z_{(t_1, t_2]}$.

We show how to obtain the outputs of Algorithm 1 from the outputs of $\mathcal{M}'$. For each time step $t \in [T]$ consider the dyadic decomposition of the interval $(0, t]$ into $k$ intervals $(t_0, t_1], (t_1, t_2], \ldots, (t_{k-1}, t_k]$, corresponding to nodes in the binary tree, where $t_0 = 0$, $t_k = t$, and $k \leq \log T + 1$. Add the outputs corresponding to the nodes in the dyadic decomposition of $(0, t]$ to obtain

$$\sum_{i \in [k]} F[t_i] - F[t_{i-1}] + Z_{(t_{i-1}, t_i]} = F[t_k] - F[0] + \sum_{i \in [k]} Z_{(t_{i-1}, t_i]} = F[t] + Z[t],$$

where the last equality holds because $Z$ is a binary-tree random variable (see Definition 2.4). The right-hand side is exactly the $t$-th output of Algorithm 1.

We now show that $\mathcal{M}'$ is $\rho$-item-level-zCDP, which implies that Algorithm 1 is $\rho$-item-level-zCDP. For each level $\ell \in [0, \log T]$ of the binary tree, define a vector $G_\ell$ of length $\frac{T}{2^\ell}$ at that level as follows:

$$G_\ell[i] = F[i \cdot 2^\ell] - F[(i-1) \cdot 2^\ell] \quad \text{for all } i \in [T/2^\ell].$$

The random variable $G_\ell[i] + Z_{(2^\ell \cdot (i-1), 2^\ell \cdot i]}$ equals the output of $\mathcal{M}'$ for node $(2^\ell \cdot (i - 1), 2^\ell \cdot i]$ in the binary tree. Let $G = (G_0, G_1 \ldots, G_{\log T})$. Mechanism $\mathcal{M}'$ corresponds to applying the Gaussian mechanism (Lemma A.10) to the output vector $G$, since the variables $Z_{(t_1, t_2]}$ corresponding to the nodes $(t_1, t_2]$ of the binary tree are independent. We now bound the $\ell_2$-sensitivity of $G$. Let $x'$ be an item-neighboring stream of $x$, and let $u \in \mathcal{U}$ be the universe element on which the two streams differ. Define $\tilde{f}'_u, F', G'_\ell$, and $G'$ for the stream $x'$ analogously to the definitions of $\tilde{f}_u, F, G_\ell$, and $G$ for stream $x$.

**Lemma B.1** ($\ell_2$-sensitivity of $G$). *For all item-neighboring streams $x$ and $x'$,*

$$\|G - G'\|_2 \leq \sqrt{8w(\log T + 1)}. \tag{2}$$

*Proof.* We first show that for all levels $\ell \in [0, \log T]$,

$$\|G_\ell - G'_\ell\|_2 \leq \sqrt{8w}.$$

Fix some $\ell \in [0, \log T]$ and $i \in [\frac{T}{2^\ell}]$. Define $i_1 = (i - 1) \cdot 2^\ell$ and $i_2 = i \cdot 2^\ell$. Since the streams $x$ and $x'$ only differ in the occurrences of element $u$, the values $G_\ell[i]$ and $G'_\ell[i]$ differ by at most 2:

$$|G_\ell[i] - G'_\ell[i]| = |\tilde{f}_u[i_2] - \tilde{f}_u[i_1] - \tilde{f}'_u[i_2] + \tilde{f}'_u[i_1]| \leq 2, \tag{3}$$

where the inequality follows from the fact that $\tilde{f}_u, \tilde{f}'_u \in \{0, 1\}^T$.

Observe that $G_\ell[i] - G'_\ell[i] \neq 0$ only if at least one of the following hold: $\tilde{f}_u[i_1] \neq \tilde{f}_u[i_2]$ or $\tilde{f}'_u[i_1] \neq \tilde{f}'_u[i_2]$. Define the flippancy of a vector $a \in \mathbb{R}^T$, denoted flip$(a)$, as the number of pairs of adjacent entries of $a$ with different values. The condition $\tilde{f}_u[i_1] \neq \tilde{f}_u[i_2]$ implies that a "flip" occurs in the vector $\tilde{f}_u$ between indices $i_1$ and $i_2$. The same holds for $\tilde{f}'_u$. By the design of Algorithm 1 (and consequently $\mathcal{M}'$), flip$(\tilde{f}_u) \leq w$ and flip$(\tilde{f}'_u) \leq w$. Additionally, all intervals $(i_1, i_2]$ for a fixed $\ell$ are disjoint. Hence, the number of intervals $i \in [\frac{T}{2^\ell}]$ such that $G_\ell[i] \neq G'_\ell[i]$ is at most $2w$. Combining this fact with Equation (3), we obtain the following upper bound on the $\ell_2$-sensitivity of $G_\ell$ for all levels $\ell \in [0, \log T]$:

$$\|G_\ell - G'_\ell\|_2^2 = \sum_{i \in [T/2^\ell]} (G_\ell[i] - G'_\ell[i])^2 \leq 2w \cdot 2^2 = 8w.$$

Combining the equalities for all levels, we obtain

$$\|G - G'\|_2^2 = \sum_{\ell \in [0, \log T]} \|G_\ell - G'_\ell\|_2^2 \le 8w(\log T + 1).$$

This concludes the proof of Lemma B.1. ∎

Recall that mechanism $\mathcal{M}'$ corresponds to applying the Gaussian mechanism to the output vector $G$. By the $\ell_2$-sensitivity bound for $G$ (Lemma B.1), and the privacy of the Gaussian mechanism (Lemma A.10), we obtain that $\mathcal{M}'$ is $(8w(\log T + 1)\rho'/2)$-zCDP, where $\rho'$ is chosen in Step 1 of Algorithm 1. By the choice of $\rho'$, mechanism $\mathcal{M}'$ (and hence, Algorithm 1) is $\rho$-item-level-zCDP.

Next, we analyze the accuracy of Algorithm 1. Suppose the input stream $x$ has maximum flippancy at most $w$. Then the variables $\tilde{f}_u$ from Algorithm 1 with input stream $x$ satisfy $\tilde{f}_u = f_u(x)$. Recall that $\mathsf{CountDistinct}(x) \in \mathbb{R}^T$ denotes the vector of distinct counts for $x$. Then $\mathsf{CountDistinct}(x) = \sum_{i \in \mathcal{U}} f_u(x) = \sum_{i \in \mathcal{U}} \tilde{f}_u(x) = s - Z$, where $s$ is the vector of outputs of Algorithm 1 defined in Step 10. As a result, $\mathsf{ERR}_{\mathsf{CountDistinct}}(x, s) = \max_{i \in [T]} |Z[t]|$. Each $Z[t]$ is the sum of at most $\log T + 1$ independent Gaussian random variables distributed as $\mathcal{N}(0, \frac{1}{\rho'})$. Therefore, $Z[t]$ is also Gaussian with mean 0 and variance at most $\frac{\log T + 1}{\rho'}$. We bound the error of our algorithm by standard concentration inequalities for Gaussian random variables. Set $m = \sqrt{16w(\log T + 1)^2/\rho}$. By Lemma F.2,

$$\Pr[\mathsf{ERR}_{\mathsf{CountDistinct}}(x, s) \ge m] = \Pr\left[\max_{t \in [T]} Z[t] \ge m\right] \le 2T e^{-\frac{m^2 \rho'}{2(\log T + 1)}} = 2T e^{-2(\log T + 1)} = \frac{2}{e^2 T}.$$

Note that $\frac{2}{e^2 T} \le \frac{1}{100}$ for large enough $T$, which concludes the proof of Theorem 2.2. ∎

## C  Proofs omitted from Section 2.2

In this section, we prove Theorem 2.1 by formalizing the ideas described at the start of Section 2.2. Then, in Section C.1, we prove Theorem 1.5 using Theorem 2.1.

The accuracy and privacy guarantees of the sparse vector technique (Algorithm 2) are stated in Theorem C.2.

**Definition C.1** ($\gamma$-accuracy [24]). *Let $(a_1, \ldots, a_k) \in \{\mathsf{Above}, \mathsf{Below}\}^k$ be a vector of answers in response to $k$ queries $q_1, \ldots, q_k$ on a dataset $x$. We say $(a_1, \ldots, a_k)$ is $\gamma$-accurate if $q_t(x) \ge -\gamma$ for all $a_t = \mathsf{Above}$ and $q_t(x) \le \gamma$ for all $a_t = \mathsf{Below}$.*

**Theorem C.2** ([24, 48]). *Algorithm 2 is $\rho$-zCDP. Let $k$ be the index of the last "$\mathsf{Above}$" query answered by Algorithm 2 (before cutoff $c$ has been crossed). For all $\beta \in (0, 1)$, with probability at least $1 - \beta$, the vector of answers to the queries $q_1, \ldots, q_k$ is $\gamma$-accurate for $\gamma = \frac{8c(\ln k + \ln(2c/\beta))}{\sqrt{2\rho}}$.*

To prove Theorem 2.1, we use a slightly stronger result (Corollary C.3) on the accuracy of Algorithm 1.

**Corollary C.3.** *Fix $\rho > 0$, sufficiently large $T \in \mathbb{N}$, and a flippancy bound $w \le T$. Algorithm 1 satisfies the following accuracy guarantee for all streams $x \in \mathcal{U}_\pm^T$ and $t \in [T]$: if at most $\ell$ elements in the prefix $x[1 : t]$ of the stream $x$ have flippancy greater than $w$, then, with probability at least $1 - \frac{1}{T}$, Algorithm 1 has error $O(\ell + \sqrt{\frac{w \log^2 T}{\rho}})$ over all time steps from 1 to $t$.*

*Proof.* The proof is similar to the accuracy analysis in Theorem 2.2, once we observe that $\mathsf{CountDistinct}(x) \le \ell \cdot 1^T + \sum_{u \in \mathcal{U}} \tilde{f}_i(x)$, where $1^T$ is a vector of length $T$. ∎

We are now ready to prove Theorem 2.1.

*Proof of Theorem 2.1.* We start by showing that Algorithm 3 is $\rho$-item-level-zCDP. Algorithm 3 accesses the stream $x$ via Algorithm 2 and the algorithms $\mathcal{B}_i$ for $i \in [0, \log T]$ (instantiations of Algorithm 1). By Theorem 2.2, Algorithm 1 with privacy parameter $\rho'$ is $\rho'$-item-level-zCDP. Since we use $(\log T + 1)$ instantiations of Algorithm 1, each with privacy parameter $\frac{\rho}{2(\log T + 1)}$, by

composition, the aggregate of the calls to Algorithm 1 is $(\frac{\rho}{2})$-item-level-zCDP. We now show that the aggregate of the calls to Algorithm 2 is $(\frac{\rho}{2})$-item-level-zCDP. Note that the queries $q_t$ for $t \in [T]$ considered in Step 11 of Algorithm 3 have sensitivity 1 for item-neighboring streams (the number of items with flippancy above a certain threshold can change by at most 1 for item-neighboring streams). By Theorem C.2, the aggregate of the calls to Algorithm 2 is $(\frac{\rho}{2})$-item-level-zCDP. Another invocation of the composition lemma gives that Algorithm 3 is $\rho$-item-level-zCDP.

We now analyze the accuracy of Algorithm 3. Set $\beta_{\mathsf{SVT}} = 0.005$, $k = T$, $c = \log T$, and $\gamma_{\mathsf{SVT}} = \frac{8 \log T (\log T + \log(400 \log T))}{\sqrt{2\rho}}$. Let $E$ be the event that the vector of answers output by the sparse vector algorithm (Algorithm 2) until the cutoff point $\log T$ is $\gamma_{\mathsf{SVT}}$-accurate. By Theorem C.2, $\Pr[E] \geq 0.996$. We condition on $E$ for most of the following proof.

Set $t_{-1}^* = 1$. Let $t_i^*$ be the last time step at which the output of instance $\mathcal{B}_i$ is used as the output of Algorithm 3. Instance $\mathcal{B}_i$ of Algorithm 1 is run with parameter $w = 2^i$. Conditioned on event $E$, its outputs are used only at times $t_{i-1}^* < t \leq t_i^*$ when at most $\ell_i = O\left(\frac{\log^2 T}{\sqrt{\rho}}\right) + \sqrt{\frac{2^i}{\rho}}$ elements have flippancy greater than $2^i$. By Corollary C.3, with probability at least $1 - \frac{1}{T}$, the error of $\mathcal{B}_i$ over time steps $t_{i-1}^*, \ldots, t_i^*$ is

$$O\left(\frac{\log^2 T + \sqrt{2^i \log^2 T}}{\sqrt{\rho}}\right).$$

Since exactly $(\log T + 1)$ instances of Algorithm 1 are run within Algorithm 3, a union bound over the failure probability of each of those instances gives us the following: Conditioned on event $E$, with probability at least $1 - \frac{\log T + 1}{T}$, the error of Algorithm 3 over time steps $t \in [T]$ is

$$O\left(\frac{\log^2 T + \sqrt{w_{\max}[t] \log^2 T}}{\sqrt{\rho}}\right). \tag{4}$$

This bound on the error holds with probability $1 - \frac{\log T + 1}{T} \geq 0.995$ for sufficiently large $T$.

**Claim C.4.** *Let $w_t$ be the (true) maximum flippancy of the sub-stream $x[1 : t]$, consisting of the first $t$ entries of the input stream $x \in \mathcal{U}_\pm^T$ to Algorithm 3. Then, for all $t \in [T]$, when the algorithm reaches Step 14,*
$$w_{\max}[t] \leq \max(2w_t, 2\rho\gamma_{\mathsf{SVT}}^2).$$

*Proof.* We consider two cases.

**(Case 1)** $t \in [T]$ during which count $< c$ for Algorithm 2.

Let $z$ be the value of $w_{\max}[t]$ when Algorithm 3 reaches Step 14. If $z = 1$ then $z = w_{\max}[t] \leq 2\gamma_{\mathsf{SVT}}^2$ since $T > 1$, $\rho < 1$. So, instead assume that $z \geq 2$. Let $t^* \leq t$ be the time step where $w_{\max}[t^*]$ is doubled from $\frac{z}{2}$ to $z$ during an execution of Step 13 of the **while** loop. This only happens if Algorithm 2 outputs "Above" for the following query:

$$\left| \left\{ u \in \mathcal{U} : \mathsf{flip}(u, x[1 : t^*]) \geq \frac{z}{2} \right\} \right| - \sqrt{\frac{z}{2\rho}}.$$

If at this point $\frac{z}{2} \leq w_{t^*}$, then $\frac{z}{2} \leq w_t$ (because $w_t \geq w_{t^*}$.) Otherwise $\frac{z}{2} > w_{t^*}$ and therefore $|\{u \in \mathcal{U} \mid \mathsf{flip}(u, x[1 : t^*]) \geq \frac{z}{2}\}| = 0$. In this case, by applying Theorem C.2, we get that $0 - \sqrt{\frac{z}{2\rho}} \geq -\gamma_{\mathsf{SVT}}$, which implies that $z \leq 2\rho\gamma_{\mathsf{SVT}}^2$.

**(Case 2)** $t \in [T]$ during which count $\geq c$ for Algorithm 2.

Suppose there is some $t \in [T]$ during which count $\geq c$. Consider the last time step $t^* \in [T]$ when Step 7 of Algorithm 2 is run (for this time step, count $= c - 1$). At this time step, $w_{max}[t^*]$ doubles from $\frac{T}{2}$ to $T$, after which it never changes again. By case (1), we have that $w_{max}[t^*] = T \leq \max(2w_{t^*}, 2\gamma_{\mathsf{SVT}}^2)$. Since for all $t \geq t^*$ it holds $w_{t^*} \leq w_t$ and $w_{max}[t] = w_{max}[t^*]$, then $w_{max}[t] \leq \max(2w_t, 2\rho\gamma_{\mathsf{SVT}}^2)$ for all $t \geq t^*$. This concludes the proof of Claim C.4. ∎

Now we substitute the upper bound on $w_{max}[t]$ from Claim C.4 into Equation (4) and apply $w_t \leq w$. We get that, for sufficiently large $T$, conditioned on event $E$, with probability at least $0.995$, the maximum error of Algorithm 3 over time steps $t \in [T]$ is

$$\mathrm{O}\left(\frac{\log^2 T + \sqrt{\max(w, 2\rho\gamma_{\mathsf{SVT}}^2)\log^2 T}}{\sqrt{\rho}}\right) = \mathrm{O}\left(\frac{\sqrt{\max\left(\log^6 T,\ 2w\log^2 T\right)}}{\sqrt{\rho}}\right). \qquad (5)$$

Finally, by a union bound over the event $E$ and the event that the error of Algorithm 3 is greater than Equation (5) we obtain: For sufficiently large $T$, the maximum error of Algorithm 3 over time steps $t \in [T]$ is bounded by Equation (5) with probability at least $0.99$. ∎

## C.1 Proof sketch of Theorem 1.5

In this section, we sketch how to complete the proof of Theorem 1.5 using Theorem 2.1 together with a result of Jain et al. [44] on mechanisms for estimating functions of sensitivity at most 1 in the continual release model.

**Theorem C.5** (Mechanism for sensitivity-1 functions [44]). *Let $f: \mathcal{U}_{\pm}^* \to \mathbb{R}$ be a function of $\ell_2$-sensitivity at most 1. Define $F: \mathcal{U}_{\pm}^T \to \mathbb{R}^T$ so that $F(x) = [f(x[1:1]), \ldots, f(x[1:T])]$. Fix $\rho \in (0,1]$ and sufficiently large $T \in \mathbb{N}$. Then, there exists a $\rho$-item-level-zCDP mechanism for estimating $F$ in the continual release model that is $\alpha$-accurate where $\alpha = \mathrm{O}\left(\min\left\{\sqrt[3]{\frac{T\log T}{\rho}}, T\right\}\right)$.*

Note that $\mathsf{CountDistinct}(x)[t]$ has $\ell_2$-sensitivity one for item-neighboring streams for all $t \in [T]$. Let $\mathcal{M}'$ be the mechanism from Theorem C.5. Then $\mathcal{M}'$ can be used for estimating $\mathsf{CountDistinct}$ under continual release for turnstile streams with the error guarantee stated in Theorem C.5. When the maximum flippancy of the stream is larger than roughly $\rho^{1/3}T^{2/3}$, the mechanism $\mathcal{M}'$ achieves better error than that of Theorem 2.1 (and it achieves worse error when the maximum flippancy of the stream is smaller than this threshold). A simple modification of Algorithm 3 can get the best of both worlds—instead of having base mechanisms $\mathcal{B}_0, \ldots, \mathcal{B}_{\log T}$ that each run Algorithm 1 with different flippancy parameters as input, we only have $k + 2 = \min(\rho^{1/3}T^{2/3}, T)$ base mechanisms $\mathcal{B}_0, \ldots, \mathcal{B}_{k+1}$. Out of these, $\mathcal{B}_0, \ldots, \mathcal{B}_k$ run Algorithm 1, whereas $\mathcal{B}_{k+1}$ runs $\mathcal{M}'$. This modified algorithm has error $O\left(\min\left(\sqrt{\frac{w}{\rho}}\operatorname{polylog} T, \sqrt[3]{\frac{T\log T}{\rho}}, T\right)\right)$. The proof is similar to the proof of Theorem 2.1, with the only difference that for analyzing the error of base mechanism $\mathcal{B}_{k+1}$ we use the error guarantee of the recompute-mechanism $\mathcal{M}'$.

Finally, Theorem 1.5 follows by invoking the conversion from zCDP to approximate DP (Lemma A.11), and setting $\rho = \frac{\varepsilon^2}{16\log(1/\delta)}$.

## D Event-level privacy lower bound

In this section, we prove Theorem 1.6, providing a strong lower bound on the parameter $\alpha$ for every $\alpha$-accurate, *event-level* differentially private mechanism for $\mathsf{CountDistinct}$ in the continual release model for turnstile streams. This lower bound is parameterized by $w$, the maximum flippancy of the input stream.

### D.1 Reduction from InnerProducts

We obtain our lower bound by showing that every mechanism for $\mathsf{CountDistinct}$ for turnstile streams can be used to obtain an algorithm with similar accuracy guarantees for $\mathsf{InnerProducts}$, the problem of estimating answers to inner product queries in the batch model. The reduction from $\mathsf{InnerProducts}$ to $\mathsf{CountDistinct}$ combines two ideas: one is the sequential embedding technique introduced by Jain et al. [44] to prove lower bounds in the continual release model and the other is a connection between the inner product of two vectors and the number of distinct elements in the concatenation of two corresponding streams. The latter idea was used by Mir et al. [50] to give lower bounds for pan-private algorithms for counting the number of distinct elements. The reduction is presented in Algorithm 4. With this reduction, we then use previously established lower bounds on accuracy for $\mathsf{InnerProducts}$ [21, 23, 50, 17] to obtain our lower bound on $\mathsf{CountDistinct}$. We start by proving Lemma D.4 (the

reduction from InnerProducts to CountDistinct). In Section D.2, we use Lemma D.4 to complete the proof of Theorem 1.6.

Algorithm 4 crucially uses the following connection between the inner product of two vectors and the number of distinct elements in the concatenation of two corresponding streams.

**Definition D.1** (Stream indicator). *For a stream $x \in \mathcal{U}_\pm^T$, let $h_x$ represent the $0/1$ vector of length $|\mathcal{U}|$, where a component $h_x[u] = 1$ iff element $u \in \mathcal{U}$ has a positive count at the end of the stream.*

**Remark D.2.** *For every pair of insertion-only streams $x$ and $x'$,*

$$\langle h_x, h_{x'} \rangle = \|h_x\|_0 + \|h_{x'}\|_0 - \|h_{x \circ x'}\|_0,$$

*where $\circ$ denotes concatenation and $\|.\|_0$ is the $\ell_0$ norm. Note that $\|h_x\|_0$ is equal to the number of distinct elements in the stream $x$.*

---

**Algorithm 4** Reduction $\mathcal{A}$ from InnerProducts to CountDistinct

    **Input:** Dataset $y = (y[1], \dots, y[n]) \in \{0, 1\}^n$, black-box access to mechanism $\mathcal{M}$ for CountDistinct in turnstile streams, and query vectors $q^{(1)}, \dots, q^{(k)} \in \{0, 1\}^n$
    **Output:** Estimates of inner products $b = (b[1], \dots, b[k]) \in \mathbb{R}^k$
1: Define the universe $\mathcal{U} = [n]$
2: Initialize stream $z^{(0)} = \perp^n$
3: **for all** $i \in [n]$ **do**
4:     If $y[i] = 1$ set $z^{(0)}[i] = +i$
5: Intialize streams $z^{(1)} = \perp^{2n}, \dots, z^{(k)} = \perp^{2n}$ and a vector $r$ of length $(2k + 1)n$
6: **for all** $(i, j) \in [n] \times [k]$ such that $q^{(j)}[i] = 1$ **do**
7:     Set $z^{(j)}[i] = +i$ and $z^{(j)}[n + i] = -i$
8: Run $\mathcal{M}$ on the stream $x \leftarrow z^{(0)} \circ z^{(1)} \circ z^{(2)} \circ \cdots \circ z^{(k)}$ and record the answers as vector $r$
9: **for all** $j \in [k]$ **do**
10:     Compute $\|q^{(j)}\|_0$ and let $b[j] = \|q^{(j)}\|_0 + r[n] - r[2jn]$
11: Return the estimates $(b[1], \dots, b[k])$

---

**Definition D.3** (Accuracy of a batch algorithm for inner products). *Let $k, n \in \mathbb{N}$. A randomized algorithm $\mathcal{A}$ is $\alpha$-accurate for InnerProducts$_{k,n}$ if, for all queries $q^{(1)}, \dots, q^{(k)} \in \{0, 1\}^n$, and all datasets $y \in \{0, 1\}^n$, it outputs $b = (b[1], \dots, b[k])$ such that*

$$\Pr_{\text{coins of } \mathcal{A}} \left[ \max_{j \in [k]} |b[j] - \langle q^{(j)}, y \rangle| \le \alpha \right] \ge 0.99.$$

We now show that if the input mechanism $\mathcal{M}$ to Algorithm 4 is accurate for CountDistinct, then Algorithm 4 is accurate for InnerProducts.

**Lemma D.4.** *Let $\mathcal{A}$ be Algorithm 4. For all $\varepsilon > 0, \delta \ge 0, \alpha \in \mathbb{R}^+$ and $n, T, k \in \mathbb{N}$, where $T \ge (2k + 1)n$, if mechanism $\mathcal{M}$ is $(\varepsilon, \delta)$-event-level-DP and $\alpha$-accurate for CountDistinct for streams of length $T$ with maximum flippancy at most $2k$, then batch algorithm $\mathcal{A}$ is $(\varepsilon, \delta)$-DP and $2\alpha$-accurate for InnerProducts$_{k,n}$.*

*Proof of Lemma D.4.* Algorithm $\mathcal{A}$ is $(\varepsilon, \delta)$-event-level-DP because $\mathcal{M}$ is $(\varepsilon, \delta)$-event-level-DP and changing a record of the dataset $y$ corresponds to changing a single entry of the stream $x$, and more specifically, an entry of the stream $z^{(0)}$ constructed in Step 4 of Algorithm 4.

We are left to prove the accuracy of $\mathcal{A}$. Fix queries $q^{(1)}, \dots, q^{(k)} \in \{0, 1\}^n$. First, observe that $z^{(0)}$ is constructed so that $y$ is its stream indicator vector. Similarly, observe that for all $j \in [k]$, the stream $z^{(j)}$ is constructed so that $q^{(j)}$ is the indicator vector for $z^{(j)}[1 : n]$, namely, the first half of $z^{(j)}$.

Next, since at time $2jn$, all of the stream entries pertaining to earlier queries $q^{(1)}, \dots, q^{(j-1)}$ have been deleted and those pertaining to $q^{(j)}$ have been inserted, $\|h_{x[1:2jn]}\|_0 = \|h_{z^{(0)} \circ z^{(j)}[1:n]}\|_0$ for $j \in [k]$.

The streams $z^{(0)}$ and $z^{(j)}[1 : n]$ for $j \in [k]$ are all insertion-only streams. By Remark D.2,

$$\langle h_{z^{(0)}}, h_{z^{(j)}[1:n]} \rangle = \|h_{z^{(0)}}\|_0 + \|h_{z^{(j)}[1:n]}\|_0 - \|h_{z^{(0)} \circ z^{(j)}[1:n]}\|_0.$$

As observed earlier, $h_{z^{(0)}} = y$, $h_{z^{(j)}[1:n]} = q^{(j)}$, and $\|h_{x[1:2jn]}\|_0 = \|h_{z^{(0)} \circ z^{(j)}[1:n]}\|_0$. Thus,

$$\langle y, q^{(j)} \rangle = \|h_{z^{(0)}}\|_0 + \|q^{(j)}\|_0 - \|h_{x[1:2jn]}\|_0. \tag{6}$$

Finally, the constructed stream $x$ has maximum flippancy at most $2k$. To see this, note that the universe elements $i \in [n]$ such that $y[i] = 1$ always have count at least 1 in $x[1:t]$ for all $t \in [(2k+1)n]$. The elements $i \in [n]$ such that $y[i] = 0$ are inserted and deleted at most once for each stream $z^{(j)}, j \in [k]$, and thus have flippancy at most $2k$ in the stream $x$.

Since the mechanism $\mathcal{M}$ for CountDistinct is $\alpha$-accurate on the constructed stream then, with probability at least 0.99, the answers of $\mathcal{M}$ are within additive error $\alpha$ of the distinct counts of the corresponding stream prefixes. Condition on this event for the rest of this proof. Then, $|r[n] - \|h_{z^{(0)}}\|_0| \leq \alpha$. Similarly, $|r[2jn] - \|h_{x[1:2jn]}\|_0| \leq \alpha$ for all $j \in [k]$. Additionally, $\|q^{(j)}\|_0$ is computed exactly by $\mathcal{A}$. Hence, by the triangle inequality, Equation 6, and the setting of $b[j]$ in Step 10, we have that $|b[j] - \langle y, q^{(j)} \rangle| \leq 2\alpha$ for all $j \in [k]$. Hence, with probability at least 0.99, all of the estimates $b[j]$ returned by $\mathcal{A}$ are within $2\alpha$ of the inner products $\langle q^{(j)}, y \rangle$, and so $\mathcal{A}$ is $2\alpha$-accurate for InnerProducts$_{k,n}$. $\blacksquare$

## D.2 From the reduction to the accuracy lower bound

In this section, we use Lemma D.4 together with a known lower bound on the accuracy of private mechanisms for answering inner-product queries to complete the proof of Theorem 1.6.

We use the following lower bound on inner product queries. Like a similar lower bound of Mir et al. [50], the proof of this theorem uses the reconstruction attacks of Dinur and Nissim [21] and Dwork et al. [23], together with the argument of De [17] that rules out reconstruction from the outputs of $(\varepsilon, \delta)$-differentially private algorithms.

**Theorem D.5** (Inner product queries lower bound (based on [21, 23, 50, 17])). *There are constants $c_1, c_2 > 0$ such that, for sufficiently large $n > 0$: if $\mathcal{A}$ is $\alpha$-accurate for InnerProducts$_{k,n}$ (Definition D.3) with $k = c_1 n$ and $\alpha = c_2 \sqrt{n}$, then $\mathcal{A}$ is not $(1, \frac{1}{3})$-differentially private.*

We first prove Theorem 1.6 for $\varepsilon = 1$ and then boost it to arbitrary $\varepsilon < 1$ using the reduction in Theorem G.1.

**Lemma D.6.** *Let $\delta \in (0, 1]$ and sufficiently large $w, T \in \mathbb{N}$ such that $w \leq T$. For all $(1, \delta)$-event-level-DP mechanisms that are $\alpha$-accurate for CountDistinct on turnstile streams of length $T$ with maximum flippancy at most $w$, if $\delta = o(\frac{1}{T})$, then*

$$\alpha = \Omega(\min(\sqrt{w}, T^{1/4})).$$

*Proof of Lemma D.6.* Fix sufficiently large $w$ such that $w \leq \sqrt{T}$. Let $c_1, c_2 > 0$ be the constants from Theorem D.5. Assume that $\mathcal{M}$ is a $\left(1, o(\frac{1}{T})\right)$-event-level-DP, $\left(\frac{c_2}{2}\sqrt{w}\right)$-accurate mechanism for CountDistinct for turnstile streams of length $T$ with maximum flippancy at most $w$. Then, set $k = \frac{w}{2}$ and $n = \frac{k}{c_1} = \frac{w}{2c_1}$. This choice of $k$ and $n$ satisfies the conditions of Lemma D.4 since the flippancy of the stream is at most $w = 2k$ and for $w \leq \sqrt{T}$ we have that $(2k+1)n = (w+1)\frac{w}{2c_1} \leq w^2 \leq T$. Therefore, $\mathcal{A}$ (Algorithm 4) is $\left(1, o(\frac{1}{T})\right)$-DP and $(c_2\sqrt{w})$-accurate for InnerProducts$_{k,n}$. Since $\frac{1}{T} \leq \frac{1}{n}$ and $w = O(n)$, we get that $\mathcal{A}$ is $(1, o(\frac{1}{n}))$-DP and $c_2\sqrt{n}$-accurate for InnerProducts$_{k,n}$, where $k = c_1 n$.

However, by Theorem D.5, $\mathcal{A}$ cannot be $(1, \frac{1}{3})$-differentially private. We have obtained a contradiction. Thus, the mechanism $\mathcal{M}$ with the desired accuracy of $O(\sqrt{w})$ does not exist. When $w = \sqrt{T}$, this argument gives a lower bound of $T^{1/4}$ on the accuracy of $\mathcal{M}$, and this lower bound applies to all larger $w$, since a mechanism that is $\alpha$-accurate for streams with maximum flippancy at most $w > w'$ is also $\alpha$-accurate for streams with maximum flippancy at most $w'$. $\blacksquare$

Finally, we invoke the reduction in Theorem G.1 to improve the dependence on $\varepsilon$ and complete the proof of Theorem 1.6.

*Proof of Theorem 1.6.* Suppose $\varepsilon < \frac{2}{T}$. For these values of $\varepsilon$, we prove an error lower bound of $\Omega(T)$, via a group privacy argument. Suppose for the sake of contradiction that $\alpha \leq T/4$. Consider universe $\mathcal{U} = [T]$. Let $x = \perp^T$ and $x'$ be a stream of length $T$ such that $x[t] = t$ for all $t \in [T]$. These data streams differ in $T$ entries. Let $r[T]$ and $r'[T]$ be the final outputs of $\mathcal{M}$ on input streams $x$ and $x'$, respectively. By the accuracy of $\mathcal{M}$, we have $\Pr[r[T] \leq T/4] \geq 0.99$. Applying Lemma A.3 on group privacy with $\varepsilon \leq 2/T$ and group size $\ell = T$, we get $\Pr[r'[T] > T/4] \leq e^2 \cdot \Pr[r[T] > T/4] + \frac{2\delta}{\varepsilon} \leq e^2 \cdot 0.01 + o(\frac{1}{T}) < 0.99$ for sufficiently large $T$. But $\mathsf{CountDistinct}(x') = T$, so $\mathcal{M}$ is not $T/4$-accurate for $x'$, a contradiction. Hence, $\alpha = \Omega(T)$.

Next, suppose $\varepsilon \geq \frac{2}{T}$. Lemma D.6 provides a lower bound of $\alpha' = \Omega\left(\min\left(\sqrt{w}, T^{1/4}\right)\right)$ for $(\varepsilon' = 1, \delta' = o(1/T))$-event-level-DP, $\alpha'$-accurate mechanisms for $\mathsf{CountDistinct}$ on turnstile streams of length $T$ with maximum flippancy at most $w$. By invoking Theorem G.1, we obtain the following lower bound on accuracy for $(\varepsilon, \delta)$-DP mechanisms where $\delta = \frac{\delta'\varepsilon}{10} = o(\frac{\varepsilon}{T})$:

$$\alpha = \frac{1}{\varepsilon}\Omega\left(\sqrt{w}, (\varepsilon T)^{1/4}\right) = \Omega\left(\min\left(\frac{\sqrt{w}}{\varepsilon}, \frac{T^{1/4}}{\varepsilon^{3/4}}\right)\right).$$

Overall, since for different parameter regimes, we get lower bounds $\Omega(T)$ and $\Omega\left(\min\left(\frac{\sqrt{w}}{\varepsilon}, \frac{T^{1/4}}{\varepsilon^{3/4}}\right)\right)$, our final result is a lower bound of $\Omega\left(\min\left(\frac{\sqrt{w}}{\varepsilon}, \frac{T^{1/4}}{\varepsilon^{3/4}}, T\right)\right)$. ∎

# E  Item-level privacy lower bound

In this section, we prove Theorem 1.7 that provides strong lower bounds on the accuracy of any *item-level* differentially private mechanism for $\mathsf{CountDistinct}$ in the continual release model for turnstile streams. This lower bound is parameterized by $w$, the maximum flippancy of the input stream.

## E.1  Reduction from Marginals

To prove our lower bounds for $\mathsf{CountDistinct}$, we reduce from the problem of approximating marginals in the batch model.

**Definition E.1** (Marginals). *The function* $\mathsf{Marginals}_{n,d} : \{0,1\}^{n \times d} \to [0,1]^d$ *maps a dataset $y$ of $n$ records and $d$ attributes to a vector $(q_1(y), \ldots, q_d(y))$, where $q_j$, called the $j^{th}$ marginal, is defined as $q_j(y) = \frac{1}{n}\sum_{i=1}^{n} y[i][j]$.*

The reduction is presented in Algorithm 5. The privacy and accuracy guarantees of our reduction are stated in Lemma E.3. In Section E.2, we use Lemma E.3 to complete the proof of Theorem 1.7.

**Overview of the reduction.** Let $\mathcal{M}$ be an $(\varepsilon, \delta)$-DP and $\alpha$-accurate mechanism for $\mathsf{CountDistinct}$ in the continual release model. We use $\mathcal{M}$ to construct a $(O(\varepsilon), O(\delta))$-DP batch algorithm $\mathcal{A}$ that is $(\frac{\alpha}{n})$-accurate for $\mathsf{Marginals}_{n,d}$. Consider a universe $\mathcal{U} = [n] \cup \{\perp\}$ for $\mathsf{CountDistinct} : \mathcal{U}_{\pm}^T \to \mathbb{N}$. The main idea in the construction (presented in Algorithm 5) is to force $\mathcal{M}$ to output an estimate of the marginals, one attribute at a time. Given a dataset $y \in \{0,1\}^{n \times d}$, the estimation of each marginal proceeds in two phases:

- In *phase one*, $\mathcal{A}$ sends element $i$ to $\mathcal{M}$ for each record $y[i]$ with a 1 in the first attribute. The answer produced by $\mathcal{M}$ at the end of *phase one* is an estimate of the sum of the first attribute of all records $y[1], \ldots, y[n]$. This can be divided by $n$ to estimate the first marginal.

- In *phase two*, $\mathcal{A}$ 'clears the slate' by sending $-i$ to $\mathcal{M}$ for each $y[i]$ with a 1 in the first attribute.

It repeats this for each attribute, collecting the answers from $\mathcal{M}$, and then outputs its estimates for the marginals. In actuality, in both phases of estimating the $j^{th}$ marginal, $\mathcal{A}$ inputs $\perp$ for each $y[i]$ that has a 0 in the $j^{th}$ attribute. This algorithm is $(O(\varepsilon), O(\delta))$-DP for $\mathsf{Marginals}_{n,d}$ since changing one record $y[i]$ in the input to the algorithm $\mathcal{A}$ will only change occurrences of a single element $i$ (to some other element $j$) in the input to the mechanism $\mathcal{M}$. Additionally, note that this reduction works

---
**Algorithm 5** Reduction $\mathcal{A}$ from Marginals to CountDistinct
---

    **Input:** Dataset $y = (y[1], \ldots, y[n]) \in \{0,1\}^{n \times d}$ and black-box access to mechanism $\mathcal{M}$ for CountDistinct in turnstile streams

    **Output:** Estimates of marginals $b = (b[1], \ldots, b[d]) \in \mathbb{R}^d$

1: Define the universe $\mathcal{U} = [n] \cup \{\bot\}$
2: Initialize streams $z^{(1)} = \bot^{2n}, \ldots, z^{(d)} = \bot^{2n}$ and a vector $r$ of length $2nd$
3: **for all** $(i,j) \in [n] \times [d]$ such that $y[i][j] = 1$ **do**
4:     Set $z^{(j)}[i] = +i$.                                               ▷ phase one
5:     Set $z^{(j)}[n+i] = -i$.                                      ▷ phase two
6: Run $\mathcal{M}$ on the stream $x \leftarrow z^{(1)} \circ z^{(2)} \circ \cdots \circ z^{(d)}$ and record the answers as vector $r$
7: **for all** $j \in [d]$ **do**
8:     $b[j] = r[(2j-1)n]/n$
9: Return estimates $(b[1], \ldots, b[d])$

---

equally well in the "likes" model where items can only be inserted when absent and deleted when present, since the stream produced in the reduction has this structure.

**Definition E.2** (Accuracy of an algorithm for marginals). *Let $\gamma \in [0,1]$ and $n, d \in \mathbb{N}$. The error* $\mathsf{ERR}_{\mathsf{Marginals}}$ *is defined as in Section 1. A batch algorithm $\mathcal{A}$ is $\gamma$-accurate for* $\mathsf{Marginals}_{n,d}$ *if for all datasets $y \in \{0,1\}^{n \times d}$,*

$$\Pr_{\text{coins of } \mathcal{A}} \left[ \mathsf{ERR}_{\mathsf{Marginals}}(y, \mathcal{A}(y)) \leq \gamma \right] \geq 0.99.$$

**Lemma E.3.** *Let $\mathcal{A}$ be Algorithm 5. For all $\varepsilon \in (0,1], \delta \geq 0, \alpha \in \mathbb{R}^+$ and $d, n, w, T \in \mathbb{N}$, where $T \geq 2dn$ and $w \geq 2d$, if mechanism $\mathcal{M}$ is $(\varepsilon, \delta)$-item-level-DP and $\alpha$-accurate for CountDistinct for streams of length $T$ with maximum flippancy at most $w$ in the continual release model, then batch algorithm $\mathcal{A}$ is $(2\varepsilon, 4\delta)$-DP and $\frac{\alpha}{n}$-accurate for* $\mathsf{Marginals}_{n,d}$.

*Proof.* We start by reasoning about privacy. Fix neighboring datasets $y$ and $y'$ that are inputs to batch algorithm $\mathcal{A}$ (Algorithm 5). (Datasets $y$ and $y'$ differ in one row.) Let $x$ and $x'$ be the streams constructed in Step 6 of $\mathcal{A}$ when it is run on $y$ and $y'$, respectively. By construction, $x$ and $x'$ are 2-item-neighbors. Since $\mathcal{M}$ is $(\varepsilon, \delta)$-item-level-DP, and $\mathcal{A}$ only post-processes the outputs received from $\mathcal{M}$, closure under post-processing (Lemma A.8) and group privacy (Lemma A.3) implies that $\mathcal{A}$ is $(2\varepsilon, 4\delta)$-DP (where we use the fact that $\frac{e^{2\varepsilon}-1}{e^{\varepsilon}-1} \leq 4$ for all $\varepsilon \in (0,1]$).

Now we reason about accuracy. Let $x = (x[1], \ldots, x[2dn])$ be the input stream provided to $\mathcal{M}$ when $\mathcal{A}$ is run on dataset $y$. Recall that $\mathsf{CountDistinct}(x)[t]$ is the number of distinct elements in stream $x$ at time $t$. By construction, for all $j \in [d]$, the $j^{\text{th}}$ marginal $q_j(y)$ of the dataset $y$ is related to $\mathsf{CountDistinct}(x)[(2j-1)n]$ as follows

$$q^{(j)}(y) = \frac{1}{n} \sum_{i \in [n]} y[i][j] = \frac{1}{n} \cdot \mathsf{CountDistinct}(x)[(2j-1)n]. \tag{7}$$

Notice that: (1) The coins of $\mathcal{A}$ are the same as the coins of $\mathcal{M}$ (since the transformation from $\mathcal{M}$ to $\mathcal{A}$ is deterministic). (2) The marginals are computed in Step 8 of Algorithm 5 using the relationship described by Equation (7). (3) The maximum flippancy of the stream constructed in Algorithm 5 is at most $2d$, since each item $i \in \mathcal{U}$ is inserted and deleted at most once in each $z^{(j)}$ for $j \in [d]$. We obtain that $\mathcal{A}$ inherits its probability of success from $\mathcal{M}$:

$$
\begin{aligned}
\Pr_{\text{coins of } \mathcal{A}} \left[ \mathsf{ERR}_{\mathsf{Marginals}_{n,d}}(y, \mathcal{A}(y)) \leq \frac{\alpha}{n} \right] &= \Pr_{\text{coins of } \mathcal{A}} \left[ \max_{j \in [d]} |q_j(y) - b[j]| \leq \frac{\alpha}{n} \right] \\
&= \Pr_{\text{coins of } \mathcal{M}} \left[ \max_{t \in \{n, \ldots, (2j-1)n\}} |\mathsf{CountDistinct}(x)[t] - r[t]| \leq \alpha \right] \\
&\geq \Pr_{\text{coins of } \mathcal{M}} \left[ \max_{t \in [T]} |\mathsf{CountDistinct}(x)[t] - r[t]| \leq \alpha \right] \\
&= \Pr_{\text{coins of } \mathcal{M}} \left[ \mathsf{ERR}_{\mathsf{CountDistinct}}(x, r) \leq \alpha \right] \geq 0.99,
\end{aligned}
$$

where we used that $\mathcal{M}$ is $\alpha$-accurate for CountDistinct for streams of length $T$ with maximum flippancy at most $w \leq 2d$. Thus, Algorithm 5 is $(\frac{\alpha}{n})$-accurate for Marginals$_{n,d}$, completing the proof of Lemma E.3. ∎

## E.2  From the reduction to the accuracy lower bound

In this section, we use Lemma E.3 (the reduction from Marginals to CountDistinct) together with previously established lower bounds for Marginals to complete the proof of Theorem D.5. The lower bounds on the accuracy of private algorithms for Marginals are stated in Items 1 and 2 of Lemma E.4 for approximate differential privacy and pure differential privacy, respectively. Item 2 in Lemma E.4 is a slight modification of the lower bound from Hardt and Talwar [38] and follows from a simple packing argument.

**Lemma E.4** (Lower bounds for Marginals [12, 38])**.** *For all $\varepsilon \in (0, 2]$ , $\delta \in [0, 1]$, $\gamma \in (0, 1)$, $d, n \in \mathbb{N}$, and algorithms that are $(\varepsilon, \delta)$-differentially private and $\gamma$-accurate for* Marginals$_{n,d}$*, the following statements hold.*

**1** [12]*. If $\delta > 0$ and $\delta = o(1/n)$, then $n = \Omega\left(\frac{\sqrt{d}}{\gamma \varepsilon \log d}\right)$.*

**2** [38]*. If $\delta = 0$, then $n = \Omega\left(\frac{d}{\gamma \varepsilon}\right)$.*

To prove Theorem 1.7, we show that the lower bound holds for $\varepsilon = 1$, and use Theorem G.1 to extend it to all $\varepsilon < 1$. The approximate-DP lower bound (on the error term $\alpha$) in Theorem 1.7 is the minimum of two terms. To prove this bound, we need to establish that, for every possible range of parameters, at least one term serves as a lower bound for $\alpha$.

**Lemma E.5.** *Let $\delta \in (0, 1]$, and sufficiently large $w, T \in \mathbb{N}$ such that $w \leq T$. For all $(1, \delta)$-item-level-DP mechanisms that are $\alpha$-accurate for* CountDistinct *on turnstile streams of length $T$ with maximum flippancy at most $w$,*

**1** *If $\delta > 0$ and $\delta = o(1/T)$, then $\alpha = \Omega\left(\min\left(\frac{\sqrt{w}}{\log w}, \frac{T^{1/3}}{\log T}\right)\right)$.*

**2** *If $\delta = 0$, then $\alpha = \Omega\left(\min\left(w, \sqrt{T}\right)\right)$.*

*Proof.* Let $\mathcal{A}$ be the algorithm for Marginals$_{n,d}$ with black-box access to an $\alpha$-accurate mechanism $\mathcal{M}$ for CountDistinct, as defined in Algorithm 5. If $T \geq 2dn$ and $w \geq 2d$, then by Lemma E.3, algorithm $\mathcal{A}$ is $(2, 4\delta)$-differentially private and $(\frac{\alpha}{n})$-accurate for Marginals$_{n,d}$. We use Lemma E.4 to lower bound $\alpha$.

**Case 1 (Approximate DP, $\delta > 0$, $\delta = o(1/n)$) :**  Suppose $w \leq T^{2/3}$. Pick number of dimensions $d = w/2$ and number of records $n = \frac{T}{w}$ (so that $T = 2dn$). If $\frac{\alpha}{n} < 1$, then by Item 1 of Lemma E.4, $n = \Omega\left(\frac{n\sqrt{d}}{\alpha \log d}\right)$ which means that $\alpha = \Omega\left(\frac{\sqrt{d}}{\log d}\right) = \Omega\left(\frac{\sqrt{w}}{\log w}\right)$. Otherwise, $\alpha \geq n \implies \alpha \geq \frac{T}{w} \geq T^{1/3} \geq \frac{T^{1/3}}{\log w} \geq \frac{\sqrt{w}}{\log w}$.

Now suppose $w = T^{2/3}$. The above argument gives a lower bound of $\Omega\left(\frac{\sqrt{T^{2/3}}}{\log T^{2/3}}\right)$ on the accuracy of $\mathcal{M}$. This lower bound applies to all $w > T^{2/3}$, since a mechanism that is $\alpha$-accurate for streams with maximum flippancy at most $w > w'$ is also $\alpha$-accurate for streams with maximum flippancy at most $w'$.

**Case 2 (Pure DP, $\delta = 0$):**  The proof for $\delta = 0$ is similar, except that we consider the cases $w \leq \sqrt{T}$ and $w > \sqrt{T}$ and use Item 2 from Lemma E.4 instead of Item 1: Suppose $w \leq \sqrt{T}$. Pick a dimension $d = w/2$, and number of entries $n = \frac{T}{w}$. If $\frac{\alpha}{n} < 1$, then by Lemma E.3 and Item 1 of Lemma E.4, $n = \Omega\left(\frac{n \cdot d}{\alpha \cdot \varepsilon}\right)$ which means that $\alpha = \Omega\left(\frac{d}{\varepsilon}\right) = \Omega(w)$. Otherwise, if $\alpha \geq n$, then $\alpha \geq \frac{T}{w} \geq \sqrt{T} \geq w$.

Now, suppose $w \geq \sqrt{T}$. Since $\mathcal{M}$ is also $\alpha$-accurate for streams of length $T$ with maximum flippancy $w' = \sqrt{T}$, the bound for $w \leq \sqrt{T}$ still applies: That is $\alpha = \Omega(w') \implies \alpha = \Omega(\sqrt{T})$.

This concludes the proof of Lemma E.5. ∎

Finally, we extend the lower bounds for $\varepsilon = 1$ from Lemma E.5 to the general case of $\varepsilon < 0.5$ using Theorem G.1.

*Proof of Theorem 1.7.* Suppose $\varepsilon < \frac{2}{T}$. For these values of $\varepsilon$, we prove an error lower bound of $\Omega(T)$, via a group privacy argument that is exactly the same as in the item-level lower bound (we direct the reader to the proof of Theorem 1.6 for more details).

Now suppose $\varepsilon \geq \frac{2}{T}$. For $\delta > 0$, Lemma E.5 provides a lower bound of $\alpha' = \Omega\left(\min\left(\frac{\sqrt{w}}{\log w}, \frac{T^{1/3}}{\log T}\right)\right)$ on accuracy for $(\varepsilon' = 1, \delta' = o(1/T))$-item-level-DP, $\alpha'$-accurate mechanisms for CountDistinct on turnstile streams of length $T$ with maximum flippancy at most $w$. By invoking Theorem G.1, we can extend this to the following lower bound for $(\varepsilon, \delta)$-DP mechanisms where $\delta = \frac{\delta'\varepsilon}{10} = o(\frac{\varepsilon}{T})$:

$$\alpha = \frac{1}{\varepsilon}\Omega\left(\min\left(\frac{\sqrt{w}}{\log w}, \frac{(\varepsilon T)^{1/3}}{\log \varepsilon T}\right)\right) = \Omega\left(\min\left(\frac{\sqrt{w}}{\varepsilon \log w}, \frac{T^{1/3}}{\varepsilon^{2/3}\log \varepsilon T}\right)\right).$$

In different parameter regimes, we get lower bounds $\Omega(T)$ and $\Omega\left(\min\left(\frac{\sqrt{w}}{\varepsilon \log w}, \frac{T^{1/3}}{\varepsilon^{2/3}\log \varepsilon T}\right)\right)$. Overall, we get a lower bound of $\Omega\left(\min\left(\frac{\sqrt{w}}{\varepsilon \log w}, \frac{T^{1/3}}{\varepsilon^{2/3}\log \varepsilon T}, T\right)\right)$. A similar proof works for $\delta = 0$. $\blacksquare$

## F  A concentration inequality for Gaussian random variables

**Lemma F.1.** *For all random variables $R \sim \mathcal{N}(0, \sigma^2)$,*

$$\Pr[|R| > \lambda] \leq 2e^{-\frac{\lambda^2}{2\sigma^2}}.$$

**Lemma F.2.** *Consider $m$ random variables $R_1, \ldots, R_m \sim \mathcal{N}(0, \sigma^2)$. Then*

$$\Pr[\max_{j \in [m]} |R_j| > \lambda] \leq 2me^{-\frac{\lambda^2}{2\sigma^2}}.$$

*Proof.* By a union bound and Lemma F.1,

$$\Pr[\max_{j \in [m]} |R_i| > \lambda] = \Pr[\exists i \in [m] \text{ such that } |R_i| > \lambda]$$

$$\leq \sum_{j=1}^{m} \Pr[|R_i| > \lambda] \leq \sum_{j=1}^{m} 2e^{-\frac{\lambda^2}{2\sigma^2}} = 2me^{-\frac{\lambda^2}{2\sigma^2}}. \qquad \blacksquare$$

## G  General lower bound reduction for small $\varepsilon$

We describe an adaptation of a folklore reduction to our problem of interest that allows us to extend a lower bound for $\varepsilon = 1$ to any $\varepsilon < 1$. The theorem is stated for item-level differential privacy, but applies to event-level differential privacy as well.

**Theorem G.1.** *Let $\alpha : \mathbb{R} \to \mathbb{R}$ be an increasing function. Let $T \in \mathbb{N}$, and $\varepsilon, \delta \in [0,1]$ such that $\varepsilon \geq \frac{1}{T}$. If for all $T \in \mathbb{N}$, every mechanism for CountDistinct that is $(1, \delta)$-item-level DP for streams of length $T$ has error parameter at least $\alpha(T)$, then every mechanism for CountDistinct that is $(\varepsilon, \delta\varepsilon/10)$-item-level DP for streams of length $T'$ has error parameter at least $\frac{\alpha(\varepsilon T')}{2\varepsilon}$.*

*Proof.* Let $T' \in \mathbb{N}$, $\varepsilon, \delta \in [0,1]$. Let $\ell = \lfloor 1/\varepsilon \rfloor$. We prove the contrapositive. Namely, let $\mathcal{M}'$ be a mechanism for CountDistinct that is $\frac{\alpha(\varepsilon T')}{2\varepsilon}$-accurate and $(\varepsilon, \frac{\delta\varepsilon}{10})$-item-level-DP for streams of length $T'$. We will use $\mathcal{M}'$ to construct a mechanism $\mathcal{M}$ for CountDistinct that is $\alpha(T)$-accurate and $(1, \delta)$-item-level-DP for streams of length $T = \lceil T'/\ell \rceil$.

Given a stream $x \in \mathcal{U}_\pm^T$ of length $T = \lceil T'/\ell \rceil$, we create a new stream $x'$ of length $T'$ with insertions and deletions from a larger universe $\mathcal{U}' = [\ell] \times \mathcal{U}$ (so every item in $\mathcal{U}$ corresponds to $\ell$ distinct items in $\mathcal{U}'$), as follows: Initialize $x'$ to be empty. For $t \in [T]$, if $x[t] = +i$, append $+(i,1), \ldots, +(i,\ell)$ to $x'$. Similarly, if $x[t] = -i$, append $-(i,1), \ldots, -(i,\ell)$. If $x[t] = \bot$ then, append $\ell$ stream entries $\bot$

to $x'$. Finally, define the output of $\mathcal{M}$ as follows: run $\mathcal{M}'$ on $x'$ and, for each time step $t \in T$, output $\mathcal{M}(x)[t] = \frac{1}{\ell} \cdot \mathcal{M}'(x')[t \cdot \ell]$.

Changing a single item of $x$ changes at most $\ell$ items of $x'$. Therefore, by the privacy guarantee of $\mathcal{M}'$ and by group privacy, $\mathcal{M}$ is $(1, \delta)$-item-level DP, for streams of length $T = \lceil T'/\ell \rceil$. Finally, the accuracy guarantee of $\mathcal{M}$ follows from the accuracy guarantee of $\mathcal{M}'$: By the construction of $x'$, it holds $\mathsf{CountDistinct}(x')[t \cdot \ell] = \ell \cdot \mathsf{CountDistinct}(x)[t]$, and by the accuracy of $\mathcal{M}'$, with probability $0.99$, the maximum error of $\mathcal{M}'$ on every input stream $x'$ of length $T'$ is at most $\alpha(\varepsilon T')/2\varepsilon$. Therefore, with probability at least $0.99$, the error of $\mathcal{M}$ is at most $\frac{1}{\ell} \cdot \alpha(\varepsilon T')/2\varepsilon \leq \frac{1}{2\varepsilon\ell} \cdot \alpha(\varepsilon \cdot \ell \cdot T) \leq \alpha(T)$, as desired. This concludes the proof of Theorem G.1. $\blacksquare$