# OpenReview forum: "Counting Distinct Elements in the Turnstile Model with Differential Privacy under Continual Observation"
_NeurIPS.cc/2023/Conference — NeurIPS 2023 poster_

### Official Review · Reviewer_wtJp · 2023-07-04

**Soundness:** 3 good
**Presentation:** 3 good
**Contribution:** 2 fair
**Rating:** 5
**Confidence:** 2

**Summary:**

The paper considers the problem of differentially private cardinality estimation under continual observation in the turnstile model. With only insertions, existing algorithms showcase the additive error is polylogT. With both insertions and deletions, this paper demonstrates that the worst case additive error is at least T^{.25}. While mapping this problem directly to the summation problem (with binary mechanism) and bounding the maximum flip with w, one can directly achieve wpolylogT additive error. This paper further reduces the factor w to sqrt(w) by using the Gaussian mechanism.

**Strengths:**

This paper studies a very important problem, distinct count with continual observation. The observation of the hardness of the turnstile model and the reason behind the maximum flip are well presented. Adaptively estimating the maximum flip with sparse vector technique is intuitive.
(After reading other reviewers opinion and rethinking what it is meant for a valuable theoretical result. I decided to increase my rating. While it may not yet to be practical, this paper considered important and hard problem and presented interesting results.)

**Weaknesses:**

When it comes to streaming, data streams can have fast velocity and huge volumes (e.g. internet traces). As a result, it motivated the design for data summaries such as flajolet-martin sketch using sublinear space to estimate distinct count of the stream. Since the paper didn’t introduce any memory-efficient algorithms for the distinct count in the turnstile model, the contribution of studying differential privacy in this setting is greatly weakened.

**Questions:**

Need more motivations. What might be an application for the proposed algorithm?

---

> ### Author Rebuttal · Authors · 2023-08-09
>
> We thank the reviewer for their careful reading of our paper. We respond to specific questions below:
>
> > Since the paper didn’t introduce any memory-efficient algorithms for the distinct count in the turnstile model, the contribution of studying differential privacy in this setting is greatly weakened.
>
> That is correct, but even the unconstrained problem was not studied previously; Prior to our work, (basically) nothing was known about cardinality estimation with deletions under continual release—with or without space constraints.  We consider only the privacy constraint, since understanding the information-theoretic limitations is a precondition to understanding the additional effect of space constraints.
>
> Furthermore, our lower bounds apply to all algorithms, including space-constrained algorithms.
>
> Space complexity is certainly a natural topic for future work.
>
> > Need more motivations. What might be an application for the proposed algorithm?
>
> Our algorithm (and future modifications) can be applied in any setting where the "elements" being counted are individuals who come and go. As we we mention in Lines 33-34, applications include "monitoring traffic on websites, the number of patients in a country’s hospitals, and the number of customers in a store."
>
> Space constraints are only a concern in some of these applications. (Further, as mentioned in the previous response, understanding the achievable accuracy without space constraints is a key step towards understanding what is possible with limited memory.)

---

> > ### Comment · Reviewer_wtJp · 2023-08-21
> >
> > Thanks a lot for the response. I have raised my score from 4 to 5.

---

### Official Review · Reviewer_MGRg · 2023-07-05

**Soundness:** 3 good
**Presentation:** 3 good
**Contribution:** 2 fair
**Rating:** 6
**Confidence:** 3

**Summary:**

This paper investigates the problem of the continual release of distinct elements subject to differential privacy in the turnstile model.

In particular, the algorithm must maintain a multi-set S containing elements from a universe U. The algorithm should support the insertion and deletion of elements. After each update, the algorithm needs to report a number which is approximately the number of distinct items in the current set S. Finally, differential privacy constraints require the list of reported numbers over the whole interaction to be private for either (1) the contribution from one item (item-level privacy) or (2) the contribution from one update (event-level privacy). The basic question is to understand the trade-off between (a) interaction length T; (b) privacy budget (eps, delta); and (c) the additive error of reports.

It is shown, via upper and lower bounds, that the additive error is characterized by the "flip number" of the stream. Namely, it is roughly the minimum w such that the contribution from one item changes by at most w times throughout the interaction.

The upper bound combines many standard tools from differential privacy, including noise-addition mechanisms, the binary-tree mechanism, and the sparse vector technique. I find the application of these tools to be standard. The lower bound is an adaptation of a previous technique.

The lower bound and upper bound nearly match for the case of item-level privacy; However this is a gap for event-level DP.

**Strengths:**

* Counting distinct elements is a basic task in streaming algorithms. Understanding the possibility of this task under privacy constraints is a natural question. This paper presents nice progress on this question.
* This paper satisfactorily shows that the hardness of the problem is tightly connected to the "flip number" quantity, which, depending on the scenario, could be much smaller than the pessimistic bound of poly(T). In this sense, this paper goes beyond the worst-case analysis usually considered in many DP theory papers.

**Weaknesses:**

* This paper does a good job of skillfully combining techniques in DP. However, the new technical meat looks limited (or maybe I was missing something)
* This paper did not consider the space complexity of algorithms, which is usually a major resource to optimize in the streaming context.

**Questions:**

NA

**Limitations:**

The limitations and possible future directions are discussed in the paper.

---

> ### Author Rebuttal · Authors · 2023-08-09
>
> We thank the reviewer for their careful reading of our paper! We respond to specific questions below:
>
> > 1. This paper does a good job of skillfully combining techniques in DP. However, the new technical meat looks limited (or maybe I was missing something)
>
> We believe that our algorithm has several genuinely new ideas in addition to using existing techniques. We would like to highlight one of them here--specifically, the idea that allows us to get from error $w$ to $\sqrt{w}$ for streams of flippancy $w$. As in previous work on insertion-only streams, our algorithm transforms the distinct elements stream into a stream of bits for which we release prefix sums using the Tree Mechanism of \[DNPR'10\] and \[CSS'11\]. A naive analysis—say, based on group privacy—would suggest that one needs to scale the tree mechanism noise up by $w$ to cover a change of flippancy $w$. However, we show that the internal structure of the tree mechanism allows one to scale the noise only by $\sqrt{w}$.
>
> Here is one indication that this analysis is nontrivial: There are optimized noise addition schemes for prefix sums (see, e.g., Denisov et al (NeurIPS 2022) and Henzinger, Upadhyay and Upadhyay (SODA 2023)) that improve quantitatively over the tree mechanism, but it is unclear if they are compatible with the same noise reduction (from $w$ to $\sqrt w$) as the tree-based scheme.
>
> Another new idea, which was picked up on by reviewer 7TW6, lies in the careful application of sparse vector. Because the maximum flippancy can have high sensitivity, we instead count the number of items with flippancy above a given threshold.
>
> Finally, the simplicity of our algorithm is a good thing! It makes it more amenable to modifications for other constraints, such as space complexity.
>
> > 2. This paper did not consider the space complexity of algorithms, which is usually a major resource to optimize in the streaming context.
>
> That is correct, but even the unconstrained problem was not studied previously; Prior to our work, (basically) nothing was known about cardinality estimation with deletions under continual release—with or without space constraints.  We consider only the privacy constraint, since understanding the information-theoretic limitations is a precondition to understanding the additional effect of space constraints.
>
> Furthermore, our lower bounds apply to all algorithms, including space-constrained algorithms.
>
> Space complexity is certainly a natural topic for future work.

---

> > ### Comment · Reviewer_MGRg · 2023-08-18
> > **Thank you**
> >
> > Thank you to the authors for answering my questions! I would like to keep my evaluation.

---

### Official Review · Reviewer_G17t · 2023-07-07

**Soundness:** 4 excellent
**Presentation:** 4 excellent
**Contribution:** 4 excellent
**Rating:** 8
**Confidence:** 4

**Summary:**

The paper studies private counting problem in continual release under the strict turnstile model. The authors focus on two neighboring definition: item-level and event-level. To gain an upper bound of error for both settings, the authors combine max flippancy, binary tree mechanism for private prefix summation and group privacy. To enforce max flippancy, the authors use SVT to privately track the flippancy. The authors also use embedding technique to obtain a lower bound which matches the upper bound for item-level privacy and most regimes of event-level privacy.

**Strengths:**

Reading the paper is a pleasure. The authors did a good job disentangling the math under the hood into plain language and clear intuition.

The main contribution of the paper is to provide the first error analysis of private counting under continual observation in the turnstile model. The derivation of the upper bound and lower bound is clear but not trivial.

The papers also list a few interesting open problems which can be good start point for future research in this area.

**Weaknesses:**

I do not find any conspicuous weakness of the manuscript.

**Questions:**

N/A

**Limitations:**

Yes, the authors adequately discuss the limitations

---

> ### Author Rebuttal · Authors · 2023-08-09
>
> We thank the reviewer for their careful reading of our paper!

---

### Official Review · Reviewer_7TW6 · 2023-07-13

**Soundness:** 3 good
**Presentation:** 3 good
**Contribution:** 3 good
**Rating:** 7
**Confidence:** 4

**Summary:**

This paper studies the problem of counting the number of distinct elements in a turnstile stream with differential privacy under continual release. For an insertion-only stream of length $T$ and $\epsilon$-DP, it was previously shown that is possible to achieve roughly $\text{polylog}(T)/\epsilon$ additive error. The results in this paper show that this is not possible in general in turnstile streams. Instead, a particular quantity of interest seems to be the flippancy number of the stream, which is the largest number of times the contribution of a single item to the distinct element count changes over the course of the stream.

For maximum flippancy $w$, this paper shows that it is possible to achieve additive error proportional to $O(\sqrt{w})$ for approximate DP, and moreover, $\Omega(\sqrt{w})$ additive error is necessary for approximate DP and $\Omega(w)$ additive error is necessary for pure DP, and the paper notes it is relatively straightforward to achieve $O(w)$ additive error for pure DP.

The approximate DP upper bound is achieved through a standard binary tree mechanism along with a sparse vector technique, where the latter is used to guess the maximum flippancy $w$ when it is not input to the stream. The lower bound is achieved through a reduction from multiple instances of private inner product queries for the event-level lower bound and through a reduction from multiple instances of a 1-way marginal query for the item-level lower bound.

**Strengths:**

+ The separation between insertion-only and turnstile streams is a nice conceptual result.
+ The usage of the sparse vector technique to privately adapt to the growth of the maximum flippancy $w$ seems creative to me.

In summary, I think the strengths significantly outweigh the following weaknesses, though I'm a bit confused about the claimed sensitivity for maximum flippancy for event-neighboring streams (see below). Hopefully this can be resolved during the discussion phase.

EDIT: As the author response in the discussion phase has resolved this point, I've increased my score from 6 to 7.

**Weaknesses:**

- The proposed algorithms fully store the entire dataset, which can often be undesirable in the streaming model.
- There is a gap between the upper and lower bounds for certain regimes of $w$ and $T$, i.e., $w\in[\sqrt{T},T^{2/3}]$.

**Questions:**

P5, L171 and P8, L349: Why does the maximum flippancy have sensitivity 1 for event-neighboring streams? I thought the example in the first paragraph of Section 1.2 shows that the flippancy can change by $\Omega(T)$.

In terms of correctness, this still seems fine since the sparse vector technique only cares about the *number* of items with flippancy more than $w$, which seems to have sensitivity 1 for both event-neighboring streams and item-neighboring streams. On the other hand, I'm confused about the differences between the event-level and item-level privacy settings and in particular, what algorithmic components can be simplified for event-level privacy.

---

> ### Author Rebuttal · Authors · 2023-08-09
>
> We thank the reviewer for the careful reading of our paper! We will do our best to incorporate the suggested clarifications.
>
> > P5, L171 and P8, L349: Why does the maximum flippancy have sensitivity 1 for event-neighboring streams? I thought the example in the first paragraph of Section 1.2 shows that the flippancy can change by $\Omega(T)$.
>
> Thank you for pointing out this mistake in the explanatory text. Indeed, in the example we provide, the maximum flippancy does change by $\Omega(T)$ for the two event-neighboring streams. The corresponding text is a remnant of a previous notion of flippancy; we will update the relevant portions accordingly. As noted by the reviewer, this does not affect our result since we are keeping track of the number of elements with flippancy above $w$, a function that has sensitivity 1 for both event and item level privacy.
>
>
> > On the other hand, I'm confused about the differences between the event-level and item-level privacy settings and in particular, what algorithmic components can be simplified for event-level privacy.
>
>
> It is not clear what major simplifications of the algorithm are possible in the event-level setting. However, the lower bounds use different reductions in the two settings. The event-level argument reduces from the problem of releasing many inner products with a fixed binary vector (for which high-accuracy solutions are ruled out by Dinur and Nissim's seminal reconstruction attacks). The item-level argument reduces from one-way marginals. The additional flexibility one has in designing item-level attacks allows increasing the range in which they apply.
>
>
>
>
> >The proposed algorithms fully store the entire dataset, which can often be undesirable in the streaming model.
>
> That is correct, but even the unconstrained problem was not studied previously; Prior to our work, (basically) nothing was known about cardinality estimation with deletions under continual release—with or without space constraints.  We consider only the privacy constraint, since understanding the information-theoretic limitations is a precondition to understanding the additional effect of space constraints.
>
> Furthermore, our lower bounds apply to all algorithms, including space-constrained algorithms.
>
> Space complexity is certainly a natural topic for future work.
>
> > There is a gap between the upper and lower bounds for certain regimes of w and T, i.e., w \in [T^{1/2},T^{2/3}].
>
> This gap between the upper and lower bounds for event-level privacy is an interesting open question. We don't have a good guess for where the true answer lies (despite having spent time on it).
>
> That said, in practice, the range of parameters for which there is a gap is quite small; in interesting settings, it seems unlikely for an element to be inserted and deleted more than $T^{1/2}$ times.

---

> > ### Comment · Reviewer_7TW6 · 2023-08-15
> >
> > Thanks for the clarification. I acknowledge receipt of the author response and have updated my review accordingly.

---

### Official Review · Reviewer_iH8C · 2023-07-26

**Soundness:** 4 excellent
**Presentation:** 4 excellent
**Contribution:** 3 good
**Rating:** 7
**Confidence:** 4

**Summary:**

This paper studies distinct elements under the turnstile continual observation model. The paper provides both upper bounds and lower bounds for event-level and item-level privacy. The error depends on the maximum flippancy, i.e., the maximum number of times that the occurrence of an element changes from positive to 0. In particular the error has sqrt dependency of the maximum flippancy, and they also show the tightness of the result.

**Strengths:**

- Distinct element is a fundamental problem in the literature of differential privacy. This paper gives an (almost) tight result for distinct element problem for the continual release model with both insertions and deletions. Previous work only studies insertion-only model, and previous techniques can only imply an error which has linear dependence on maximum flippancy. This paper open the black box of the binary tree mechanism and shows a sqrt(maximum flippancy) dependence.

- I feel the lower bounds are very interesting. To the best of my knowledge, this is the first work formally studying the necessity of the dependence on maximum flippancy for the distinct element problem under turnstile continual observation model.

- The proposed algorithm also works for item-level DP, i.e., even when considering the neighboring stream which is obtained by removing all occurrence of one item, the algorithm is still DP.

**Weaknesses:**

- The algorithm does not work in the low memory regime, i.e., the space required is linear in the stream size.

- The lower bound of event level DP is not tight.

- For the upper bound, the algorithmic idea is not very impressive. It is basically a clever combination of several standard techniques and ideas appeared in the previous DP continual observation literature. But maybe it is not really a weakness because the lower bound shows that such clean and simple algorithm is the best one can get.

**Questions:**

What's authors' conjectured lower bound for the case when maximum flippancy is between T^{0.5} and T^{2/3}?

**Limitations:**

Authors adequately addressed the limitations.

---

> ### Author Rebuttal · Authors · 2023-08-09
>
> Thank you for your careful reading of our paper! We respond to specific questions below:
>
> > The algorithm does not work in the low memory regime, i.e., the space required is linear in the stream size.
>
> That is correct, but even the unconstrained problem was not studied previously; Prior to our work, (basically) nothing was known about cardinality estimation with deletions under continual release—with or without space constraints.  We consider only the privacy constraint, since understanding the information-theoretic limitations is a precondition to understanding the additional effect of space constraints.
>
> Furthermore, our lower bounds apply to all algorithms, including space-constrained algorithms.
>
> Space complexity is certainly a natural topic for future work.
>
> > The lower bound of event level DP is not tight. [...]
> >
> >What's authors' conjectured lower bound for the case when maximum flippancy is between T^{0.5} and T^{2/3}?
>
> This gap between the upper and lower bounds for event-level privacy is an interesting open question. We don't have a good guess for where the true answer lies (despite having spent time on it).
>
> That said, in practice, the range of parameters for which there is a gap is quite small; in interesting settings, it seems unlikely for an element to be inserted and deleted more than $T^{1/2}$ times.
>
> > For the upper bound, the algorithmic idea is not very impressive. [...] But maybe it is not really a weakness because the lower bound shows that such clean and simple algorithm is the best one can get.
>
> We believe that our algorithm has several genuinely new ideas in addition to using existing techniques. We would like to highlight one of them here--specifically, the idea that allows us to get from error $w$ to $\sqrt{w}$ for streams of flippancy $w$. As in previous work on insertion-only streams, our algorithm transforms the distinct elements stream into a stream of bits for which we release prefix sums using the Tree Mechanism of \[DNPR'10\] and \[CSS'11\]. A naive analysis—say, based on group privacy—would suggest that one needs to scale the tree mechanism noise up by $w$ to cover a change of flippancy $w$. However, we show that the internal structure of the tree mechanism allows one to scale the noise only by $\sqrt{w}$.
>
> Here is one indication that this analysis is nontrivial: There are optimized noise addition schemes for prefix sums (see, e.g., Denisov et al (NeurIPS 2022) and Henzinger, Upadhyay and Upadhyay (SODA 2023)) that improve quantitatively over the tree mechanism, but it is unclear if they are compatible with the same noise reduction (from $w$ to $\sqrt w$) as the tree-based scheme.
>
> Another new idea, which was picked up on by reviewer 7TW6, lies in the careful application of sparse vector. Because the maximum flippancy can have high sensitivity, we instead count the number of items with flippancy above a given threshold.
>
> Finally, the simplicity of our algorithm is a good thing! It makes it more amenable to modifications for other constraints, such as space complexity.

---

> > ### Comment · Reviewer_iH8C · 2023-08-21
> >
> > Thank you to the authors for the detailed responses! I would like to keep my evaluation.

---

### Decision · Program_Chairs · 2023-09-21

**Decision:**

Accept (poster)

**Comment:**

This paper studies a fundamental problem in differential privacy, and provides interesting results (both upper and lower bounds). The main limitation of the this paper is that the techniques are not very novel; nevertheless, after discussion with the reviewers, there is an agreement that the paper would be a good addition to NeurIPS.